# REDD1 promotes obesity-induced metabolic dysfunction via atypical NF-κB activation

Dong-Keon Lee ®[1,7], Taesam Kim[1,7], Junyoung Byeon ®[2], Minsik Park[1], Suji Kim[1], Joohwan Kim[1], Seunghwan Choi[1], Gihwan Lee[3], Chanin Park[3], Keun Woo Lee[4], Yong Jung Kwon[5], Jeong-Hyung Lee[2], Young-Guen Kwon[6] & Young-Myeong Kim ®[1] ✉

Regulated in development and DNA damage response 1 (REDD1) expression is upregulated in response to metabolic imbalance and obesity. However, its role in obesity-associated complications is unclear. Here, we demonstrate that the REDD1–NF-κB axis is crucial for metabolic inflammation and dysregulation. Mice lacking *Redd1* in the whole body or adipocytes exhibited restrained diet-induced obesity, inflammation, insulin resistance, and hepatic steatosis. Myeloid *Redd1*-deficient mice showed similar results, without restrained obesity and hepatic steatosis. *Redd1*-deficient adipose-derived stem cells lost their potential to differentiate into adipocytes; however, REDD1 overexpression stimulated preadipocyte differentiation and proinflammatory cytokine expression through atypical IKK-independent NF-κB activation by sequestering IκBα from the NF-κB/IκBα complex. REDD1 with mutated Lys$^{219/220}$Ala, key amino acid residues for IκBα binding, could not stimulate NF-κB activation, adipogenesis, and inflammation in vitro and prevented obesity-related phenotypes in knock-in mice. The REDD1-atypical NF-κB activation axis is a therapeutic target for obesity, meta-inflammation, and metabolic complications.

Obesity is not just a state of excess body fat or body mass, but a risk factor for the development of metabolic disorders, such as cardiovascular disease, type 2 diabetes (T2D), and immune-mediated disorders[1,2]. Obesity is associated with adipocyte hyperplasia and hypertrophy, which facilitate lipid storage and increase the population of immune cells in adipose tissues, in addition to promoting meta-inflammation and insulin resistance.

Brown adipose tissue, found primarily in human neonates, regulates body temperature through non-shivering thermogenesis; by contrast, mammalian white adipose tissue (WAT) contributes to the pathogenesis of metabolic disorders in individuals with obesity. The pathological state of obesity highly correlates with the circulating

levels of adipocyte-specific adipokines, such as adiponectin, leptin, resistin, and visfatin, and NF-κB-responsive proinflammatory cytokines, such as tumor-necrosis factor-α (TNF-α), interleukin (IL)−1β, IL-6, and monocyte chemoattractant protein-1 (MCP-1). These soluble factors play important roles in the link between obesity and T2D[3,4]. Although WATs primarily release proinflammatory cytokines, adipocytes are not the main source of circulating cytokines. Consistently, WATs contain a large number of resident macrophages and promote the recruitment of monocytes/macrophages through chemokine signaling, mediated by MCP-1 released from hypertrophic adipocytes and resident macrophages. Both types of cells are major proinflammatory mediators in obese adipose tissue[5,6]. This crosstalk between

[1]Department of Molecular and Cellular Biochemistry, School of Medicine, Kangwon National University, Chuncheon 24341, Republic of Korea. [2]Department of Biochemistry, College of Natural Sciences, Kangwon National University, Chuncheon 24341, Republic of Korea. [3]Division of Life Sciences, Division of Applied Life Science, Gyeongsang National University, Jinju 52828, Republic of Korea. [4]Division of Life Sciences, Department of Bio & Medical Big Data (BK4 Program), Gyeongsang National University, Jinju 52828, Republic of Korea. [5]GILO Foundation, Seoul 06668, South Korea. [6]Department of Biochemistry, College of Life Science and Biotechnology, Yonsei University, Seoul 03722, Republic of Korea. [7]These authors contributed equally: Dong-Keon Lee, Taesam Kim. ✉e-mail: ymkim@kangwon.ac.kr

adipocytes and macrophages perpetuates a vicious cycle of macrophage recruitment and NF-κB-dependent production of proinflammatory cytokines.

Proinflammatory cytokine production through NF-κB activation is critical in the pathogenesis of insulin resistance and T2D[7]. However, the role of NF-κB activation in adipogenesis and inflammation-mediated insulin resistance is complex and unclear. For example, genetic or pharmacological inhibition of IκB kinase β (IKKβ), a primary and canonical upstream mediator of the NF-κB pathway in innate immunity, can restore weight gain and insulin sensitivity in mice with high-fat diet (HFD)-induced obesity[8–11], suggesting that NF-κB activation is essential for obesity-induced inflammation and insulin resistance. However, contradictory reports suggest that the constitutive activation of IKKβ or adipocyte-specific IKKβ deletion can promote HFD-induced adipogenesis and insulin resistance[12,13]. In addition, palmitate, a free fatty acid known as a key mediator of obesity-induced inflammation, induces the production of the proinflammatory cytokine TNF-α without activating the canonical NF-κB pathway[14], indicating that although NF-κB activation is required for obesity-induced inflammation, it may be independent of canonical IKKβ-dependent IκBα degradation. However, alternative mechanisms for obesity-induced NF-κB activation are unknown.

Emerging evidence suggests that the expression of <u>r</u>egulated in development and <u>D</u>NA damage response 1 (REDD1) is upregulated in the skeletal muscle and liver of mice and humans with obesity[15,16]. REDD1 performs two distinct functions, i.e., it acts as an endogenous mTORC1 inhibitor and NF-κB activator[17,18]. Consistently, REDD1 atypically activates NF-κB and stimulates proinflammatory cytokine gene expression in various pathogenic conditions[18,19]. However, little is known regarding its role in meta-inflammation and insulin resistance in

conjunction with obesity. Here we explored the pathogenic role of REDD1 in obesity-induced inflammation and metabolic complications. Our findings reveal that REDD1 plays a crucial role in atypical NF-κB activation and links adipogenesis, meta-inflammation, and insulin resistance.

## Results
### *Redd1* loss ameliorates HFD-induced adipogenesis and weight gain

Consistent with the results that metabolic dysregulation elevates REDD1 expression[16,20], we found that REDD1 expression increased in the epididymal WAT (eWAT), skeletal muscle, and liver of *ob/ob*, *db/db*, and HFD-fed C57BL/6 mice compared with those in mice fed normal chow (NC) (Supplementary Fig. 1a). In addition, REDD1 expression also increased in adipocytes, stromal vascular fraction (SVF) cells, and adipose tissue macrophages isolated from eWAT of HFD-fed C57BL/6 mice (Supplementary Fig. 1b). Therefore, we investigated the role of REDD1 as a regulator of adipogenesis and obesity using *Redd1*$^{-/-}$ mice, which were confirmed by depletion of REDD1 expression in the eWAT, skeletal muscle, and liver after being fed HFD (Supplementary Fig. 1c). Under HFD conditions, obesity-related phenotypes, such as an increase in body weight, eWAT and inguinal WAT (iWAT) mass, and body fat percentage, were ameliorated without altering food intake in *Redd1*$^{-/-}$ mice compared with their wild-type (WT) littermate controls; however, these characteristics were not significantly different between the NC-fed groups (Fig. 1a, b and Supplementary Fig. 2a, b). Consistently, in *Redd1*$^{-/-}$ mice, we observed a significant decrease in the levels of plasma adipokines, including leptin and resistin, and a significant increase in the levels of the anti-inflammatory adipokine adiponectin, compared with those in their WT littermates

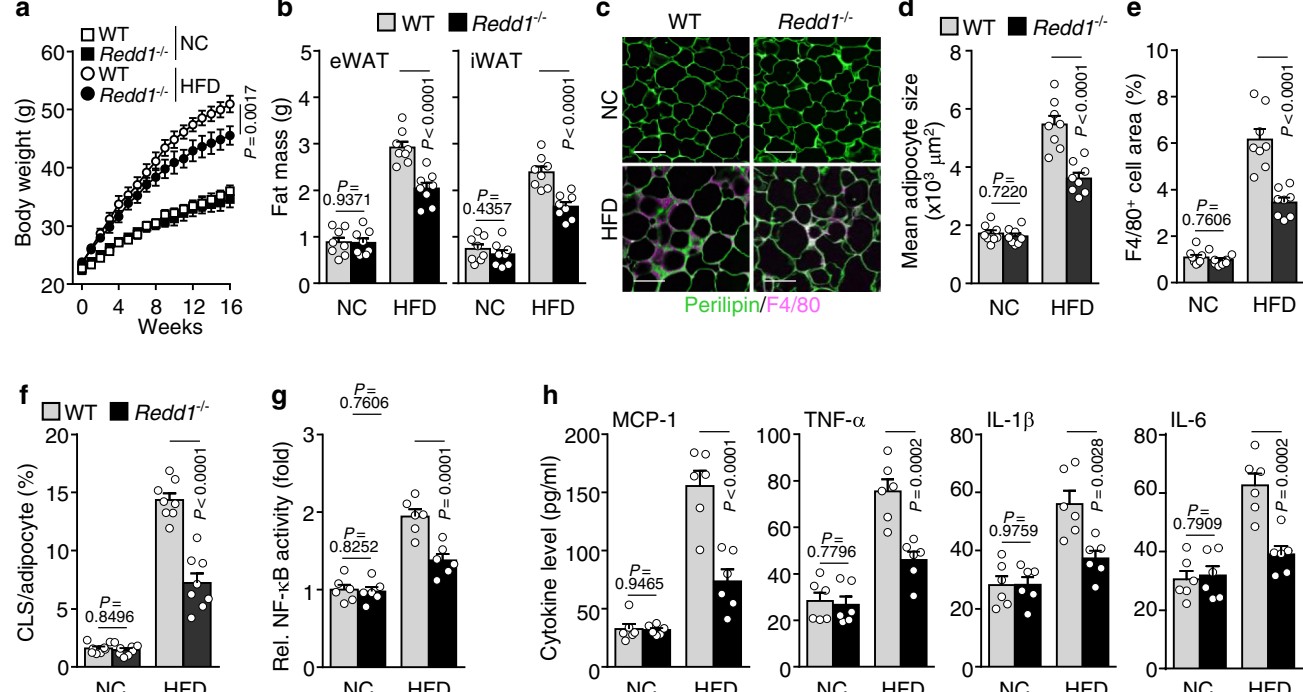

**Fig. 1 | *Redd1*$^{-/-}$ mice are protected against HFD-induced weight gain and adipose tissue expansion. a** Weight gain in *Redd1*$^{-/-}$ mice and their WT littermates fed NC or HFD for 16 weeks (*n* = 6 per group). **b** Mass of eWAT and iWAT in NC- or HFD-fed *Redd1*$^{-/-}$ mice and their WT littermates (*n* = 8 per group). **c** Representative images of perilipin (green) and F4/80 (purple) staining in the eWAT of NC- or HFD-fed *Redd1*$^{-/-}$ mice and WT littermates (*n* = 8 per group). Scale bar, 100 μm. **d** Average adipocyte size in the eWAT of NC- or HFD-fed *Redd1*$^{-/-}$ mice and WT littermates (*n* = 8 per group). **e** Relative area of F4/80-positive cells in the eWAT of

NC- or HFD-fed *Redd1*$^{-/-}$ mice and WT littermates (*n* = 8 per group). **f** Relative number of crown-like structures (CLSs) in the eWAT of NC- or HFD-fed *Redd1*$^{-/-}$ mice and WT littermates (*n* = 8 per group). **g** NF-κB activity in the eWAT from NC- or HFD-fed *Redd1*$^{-/-}$ mice and WT littermates (*n* = 6 per group). **h** Plasma levels of inflammatory cytokines in NC- or HFD-fed *Redd1*$^{-/-}$ mice and WT littermates (*n* = 6 per group). Bar graphs represent mean ± s.e.m. Statistical significance was calculated using two-way ANOVA followed by the Holm–Sidak post hoc test. Source data are provided as a Source Data file.

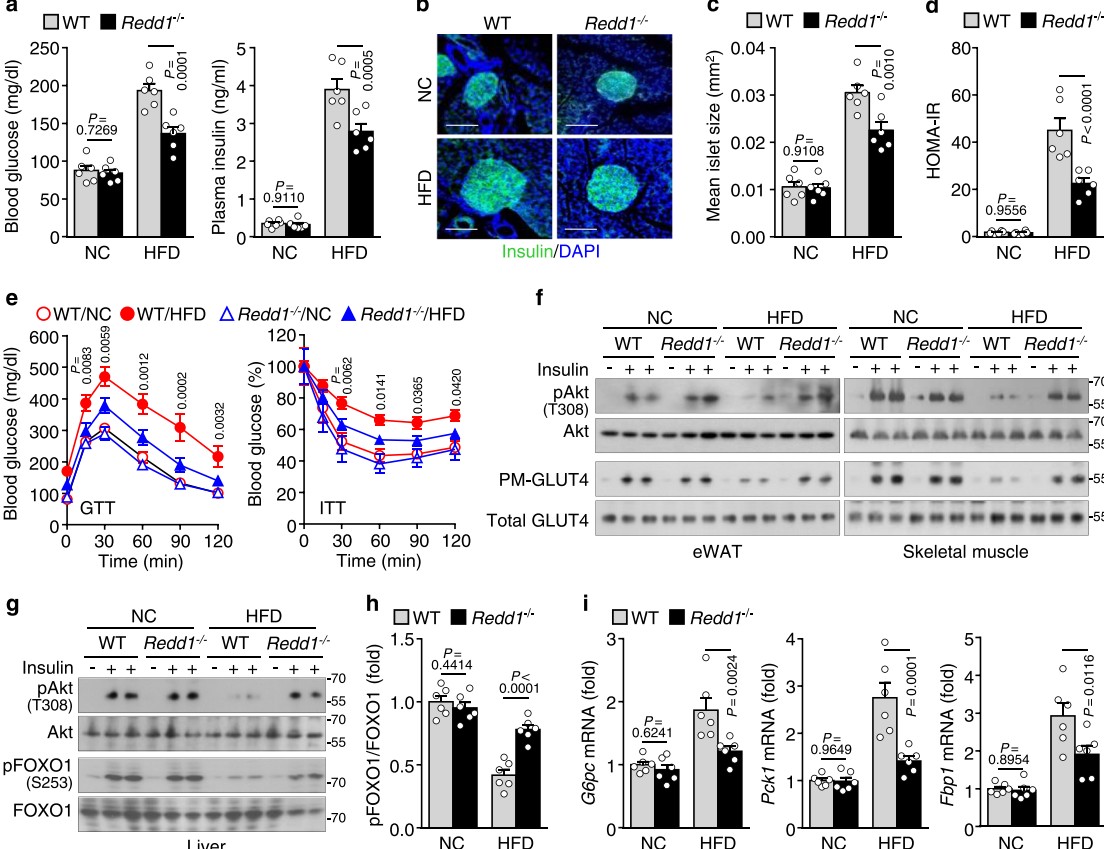

**Fig. 2 | *Redd1*⁻/⁻ mice are protected against HFD-induced metabolic dysregulation. a** Fasting plasma levels of glucose and insulin in NC- or HFD-fed *Redd1*⁻/⁻ mice and their WT littermates (*n* = 6 per group). **b** Representative images of insulin (green)-stained pancreatic islets from NC- or HFD-fed *Redd1*⁻/⁻ mice and WT littermates (*n* = 6 per group). Scale bar, 100 µm. **c** Quantification of average islet size (*n* = 6 per group). **d** Calculation of the HOMA-IR scores (*n* = 6 per group). **e** Assessment of GTT and ITT in mice fasting for 12 and 6 h, respectively, in NC- or HFD-fed *Redd1*⁻/⁻ mice and WT littermates (*n* = 8 per group). **f** Representative western blots of Akt phosphorylation and plasma membrane-associated GLUT4 (PM-GLUT4) in eWAT and skeletal muscle from mice injected i.p. with saline or insulin (*n* = 3). **g** Representative western blots of phosphorylated Akt and FOXO1 in the liver of mice injected with saline or insulin (*n* = 6). **h** Quantification of the phosphorylated FOXO1 to total FOXO1 ratio (*n* = 6 per group). **i** Quantification of *G6pc*, *Pck1*, and *Fbp1* mRNA levels in the liver (*n* = 6 per group). Bar graphs represent mean ± s.e.m. Statistical significance was calculated using two-way ANOVA followed by the Holm−Sidak post hoc test. Source data are provided as a Source Data file.

(Supplementary Fig. 2c−e), suggesting that REDD1 promotes adipogenesis. Therefore, we evaluated whether HFD-induced REDD1 regulates the expression of core adipogenic genes peroxisome proliferator-activated receptor γ (*Pparg*) and CCAAT/enhancer-binding protein α (*Cebpa*) and their downstream adipogenic target adipocyte protein 2 (*aP2*). The expression levels of these genes were significantly lower in the eWAT of *Redd1*⁻/⁻ mice than in their WT littermates under HFD conditions (Supplementary Fig. 2f−h). These data suggest that REDD1 plays an important role in HFD-induced adipogenesis.

### *Redd1* loss restrains obesity-induced inflammation

Because adipose tissue inflammation plays a crucial role in the development of insulin resistance, we investigated the possible involvement of REDD1 in adipocyte hypertrophy, macrophage infiltration, and cytokine expression in obese mice. Histological analysis revealed a significant decrease in adipocyte size (hypertrophy), with a reduction in the macrophage-specific marker F4/80-positive cell population and crown-like structure (CLS) formation, in the eWAT of HFD-fed *Redd1*⁻/⁻ mice compared with their WT littermates; however, these distinctive characteristics were not observed in the mouse groups fed NC (Fig. 1c−f). Because crosstalk between adipocytes and macrophages induces and exacerbates the initiation of NF-κB-dependent meta-inflammation in obese adipose tissue[21], we next

compared NF-κB activation and inflammatory cytokine gene expression in the eWAT of *Redd1*⁻/⁻ mice and their WT littermates. Under HFD conditions, NF-κB activity and *Ccl2, Tnfa, Il1b, and Il6* mRNA levels increased in the eWAT of *Redd1*⁻/⁻ mice and their WT littermates but were much lower in *Redd1*⁻/⁻ mice than in WT littermates (Fig. 1g and Supplementary Fig. 3a). Consistently, plasma levels of MCP-1, TNF-α, IL-1β, and IL-6 were also lowered in *Redd1*⁻/⁻ mice on HFD (Fig. 1h). In addition, under HFD conditions, levels of phospho-Stat3 and *Socs3*, known as downstream targets of IL-6 and negative regulators of insulin signal[22], were significantly lowered in the eWAT, liver, and skeletal muscle of *Redd1*⁻/⁻ mice compared with those in WT littermates (Supplementary Fig. 3b, c). Collectively, these results suggest that under the HFD challenge, REDD1 promotes macrophage infiltration into adipose tissues, leading to meta-inflammation.

### *Redd1* loss suppresses glucose metabolism

Because meta-inflammation is a crucial link between obesity and T2D[3], we examined the role of REDD1 in the pathogenesis of glucose metabolism dysregulation and T2D. Fasting blood glucose and plasma insulin levels were elevated in both *Redd1*⁻/⁻ mice and their WT littermates after 16 weeks of HFD intake; however, the levels were lower in *Redd1*⁻/⁻ mice than in WT littermates (Fig. 2a). Next, we examined insulin expression and islet size using immunohistochemistry.

Pancreatic insulin content per pixel of the islet was similar between $Redd1^{-/-}$ mice and their WT littermates fed NC or HFD (Fig. 2b). Mean islet size increased in HFD-fed $Redd1^{-/-}$ mice and their WT littermates compared with that in NF-fed mouse groups; however, it was smaller in $Redd1^{-/-}$ mice than in WT littermates (Fig. 2b, c). HOMA-IR, an indicator of insulin resistance, was found to increase in both mouse groups under HFD conditions, but significantly decrease in $Redd1^{-/-}$ mice (Fig. 2d), suggesting an improvement in insulin sensitivity and resistance in $Redd1^{-/-}$ mice. Next, the glucose tolerance test (GTT) and insulin tolerance test (ITT) were performed to characterize the effects of $Redd1$ deficiency on systemic glucose homeostasis and insulin sensitivity. Under HFD conditions, $Redd1^{-/-}$ mice had significantly improved glucose tolerance and insulin resistance compared with WT littermates (Fig. 2e). Because insulin signaling is impaired in meta-inflammation, we assayed the insulin signaling pathways in metabolic organs. Under HFD conditions, insulin-mediated phosphorylation of Akt at $Thr^{308}$ (active form) and the insulin-responsive glucose transporter GLUT4 translocation to the plasma membrane in adipose tissues and skeletal muscle reduced markedly in control WT mice but were preserved in $Redd1^{-/-}$ mice (Fig. 2f). Similarly, insulin-mediated phosphorylation of Akt at $Thr^{308}$ and FOXO1 at $Ser^{253}$ ($Ser^{256}$ in humans, inactive from) decreased in the liver of HFD-fed WT mice but were also largely preserved in $Redd1^{-/-}$ mice fed HFD (Fig. 2g, h), suggesting that $Redd1^{-/-}$ suppresses the transcriptional activity of FOXO1 in the HFD group. Consequently, the expression of the gluconeogenic genes, glucose-6-phosphatase catalytic subunit ($G6pc$), phosphoenolpyruvate carboxykinase 1 ($Pck1$), and fructose-bisphosphatase 1 ($Fbp1$), significantly decreased in the liver of $Redd1^{-/-}$ mice compared with that in their WT littermates in the HFD group (Fig. 2i), indicating that $Redd1^{-/-}$ reduces hepatic gluconeogenesis. These results suggest that obesity-induced REDD1 expression impairs insulin sensitivity and glucose metabolism by suppressing insulin signaling through increased meta-inflammation.

## Adipocyte $Redd1$ deficiency reduces weight gain and meta-inflammation

Because both adipocytes and macrophages play important roles in obesity-induced inflammation and metabolic dysregulation, we examined the role of adipocyte REDD1 in HFD-induced adipogenesis and inflammation using $Redd1^{\Delta Adipoq}$ mice, in which REDD1 expression was specifically deleted in mature adipocytes but not in SVF cells, liver tissues, and skeletal muscles under HFD conditions (Supplementary Fig. 1d). Similar to $Redd1^{-/-}$ mice, HFD-fed $Redd1^{\Delta Adipoq}$ mice exhibited decreased weight gain and eWAT and iWAT mass compared with HFD-fed control $Redd1^{fl/fl}$ mice; however, these distinctive characteristics were not observed in NC-fed groups (Fig. 3a, b and Supplementary Fig. 4a, b). The expression levels of $Pparg$, $Cebpa$, and $aP2$ were lower in the eWAT of HFD-fed $Redd1^{\Delta Adipoq}$ mice than in that of HFD-fed $Redd1^{fl/fl}$ mice (Supplementary Fig. 4c). Consistent with adipose tissue expansion, the levels of plasma leptin and resistin were lower and those of adiponectin were higher in $Redd1^{\Delta Adipoq}$ mice than in $Redd1^{fl/fl}$ mice (Supplementary Fig. 4d). Moreover, adipocyte size and F4/80$^+$ macrophage populations decreased significantly in the eWAT of $Redd1^{\Delta Adipoq}$ mice fed HFD compared with their control counterparts; however, these were similar in both groups when mice were fed NC (Fig. 3c and Supplementary Fig. 4e–g). NF-κB activation in eWAT and plasma MCP-1, TNF-α, IL-1β, and IL-6 levels were lower in HFD-fed $Redd1^{\Delta Adipoq}$ mice than in HFD-fed $Redd1^{fl/fl}$ mice, but not different between NC-fed groups (Fig. 3d–f and Supplementary Fig. 4h, i). These results suggest that adipocyte REDD1 contributes to HFD-induced obesity and meta-inflammation.

## Adipocyte $Redd1$ deficiency prevents metabolic dysregulation

We investigated the role of adipocyte REDD1 in obesity-induced glucose metabolism and insulin resistance. Consistent with findings in $Redd1^{-/-}$ mice (Fig. 1), fasting blood glucose and plasma insulin levels were significantly lower in $Redd1^{\Delta Adipoq}$ mice than in their control counterparts fed HFD (Fig. 3g and Supplementary Fig. 5a). These protective effects were confirmed by GTT and ITT, respectively, showing that $Redd1^{\Delta Adipoq}$ mice presented lower blood glucose levels after glucose injection and higher insulin-mediated glucose clearance than $Redd1^{fl/fl}$ mice fed HFD (Fig. 3h and Supplementary Fig. 5b). Consistently, insulin-mediated phosphorylation levels of the insulin receptor substrate-1 (IRS-1) at $Tyr^{895}$ and Akt at $Thr^{308}$ in eWAT and skeletal muscle and Akt at $Thr^{308}$ and FOXO1 at $Ser^{253}$ in the liver decreased in HFD-fed $Redd1^{fl/fl}$ mice; however, these remained similar to control levels in HFD-fed $Redd1^{\Delta Adipoq}$ mice (Fig. 3i, j), suggesting that insulin signaling is maintained in $Redd1^{\Delta Adipoq}$ mice. Due to FOXO1 phosphorylation, levels of $G6pc$, $Pck1$, and $Fbp1$ expression were reduced in the liver of HFD-fed $Redd1^{\Delta Adipoq}$ mice compared with those in HFD-fed $Redd1^{fl/fl}$ mice, but not different between NC-fed groups (Fig. 3k and Supplementary Fig. 5c), probably leading to the reduction of hepatic gluconeogenesis in $Redd1^{\Delta Adipoq}$ mice on HFD. These results suggest that HFD-induced REDD1 expression in adipocytes induces insulin resistance and metabolic dysregulation by stimulating meta-inflammation.

## Myeloid $Redd1$ deficiency prevents meta-inflammation without affecting adipogenesis

We examined the role of macrophage $Redd1$ in obesity-related pathological characteristics using $Redd1^{\Delta LysM}$ mice. When fed HFD, weight gain; eWAT and iWAT mass; $Pparg$, $Cebpa$, and $aP2$ expression; and plasma leptin and resistin levels increased, but adiponectin levels decreased in both $Redd1^{fl/fl}$ and $Redd1^{\Delta LysM}$ mice compared with those in NC-fed groups; however, these were not significantly different between $Redd1^{fl/fl}$ and $Redd1^{\Delta LysM}$ mice under the same dietary conditions (Fig. 4a, b and Supplementary Fig. 6a–f). This suggests that myeloid REDD1 does not affect adipocyte differentiation and adipogenesis. Consistently, adipocyte size increased similarly in both HFD-fed $Redd1^{fl/fl}$ and $Redd1^{\Delta LysM}$ mice compared with that in NC-fed mouse groups (Fig. 4c and Supplementary Fig. 6g, h). However, the population of F4/80$^+$ macrophages in the eWAT was elevated in both HFD-fed $Redd1^{fl/fl}$ and $Redd1^{\Delta LysM}$ compared with that in NC-fed mouse groups, but significantly lower in $Redd1^{\Delta LysM}$ than in $Redd1^{fl/fl}$ mice (Fig. 4c and Supplementary Fig. 6g, i). As a result, under HFD conditions, NF-κB activation in eWAT and plasma MCP-1, TNF-α, and IL-1β levels increased in both HFD-fed $Redd1^{fl/fl}$ and $Redd1^{\Delta LysM}$ mice compared with those in NC-fed mouse groups but were much lower in $Redd1^{\Delta LysM}$ mice than in $Redd1^{fl/fl}$ mice (Fig. 4d, e and Supplementary Fig. 6j, k). These data suggest that myeloid REDD1 is essential for macrophage infiltration and meta-inflammation, without affecting adipogenesis, in the context of diet-induced obesity.

## Myeloid $Redd1$ deficiency improves insulin resistance and metabolic dysregulation

We investigated the role of myeloid REDD1 in glucose metabolism. HFD-fed $Redd1^{\Delta LysM}$ and $Redd1^{fl/fl}$ mice had increased fasting blood glucose and plasma insulin levels compared with NF-fed groups; however, the levels were lower in $Redd1^{\Delta LysM}$ than in $Redd1^{fl/fl}$ mice (Fig. 4f and Supplementary Fig. 7a). Consequently, HOMA-IR index scores were lower in $Redd1^{\Delta LysM}$ than in $Redd1^{fl/fl}$ mice when fed only HFD (Fig. 4g and Supplementary Fig. 7b). Glucose tolerance and insulin resistance were significantly improved in HFD-fed $Redd1^{\Delta LysM}$ mice compared with those observed in HFD-fed control mice, but not different between NC-fed groups, as determined by GTT and ITT, respectively (Fig. 4h and Supplementary Fig. 7c). These findings suggest that myeloid REDD1 plays an important role in regulating glucose homeostasis and insulin sensitivity. Next, we examined the role of myeloid $Redd1$ deficiency in insulin signaling and gluconeogenic gene expression. HFD intake led to reduced insulin-responsive phosphorylation of IRS-1 at $Tyr^{895}$ and Akt at $Thr^{308}$ in the eWAT and skeletal

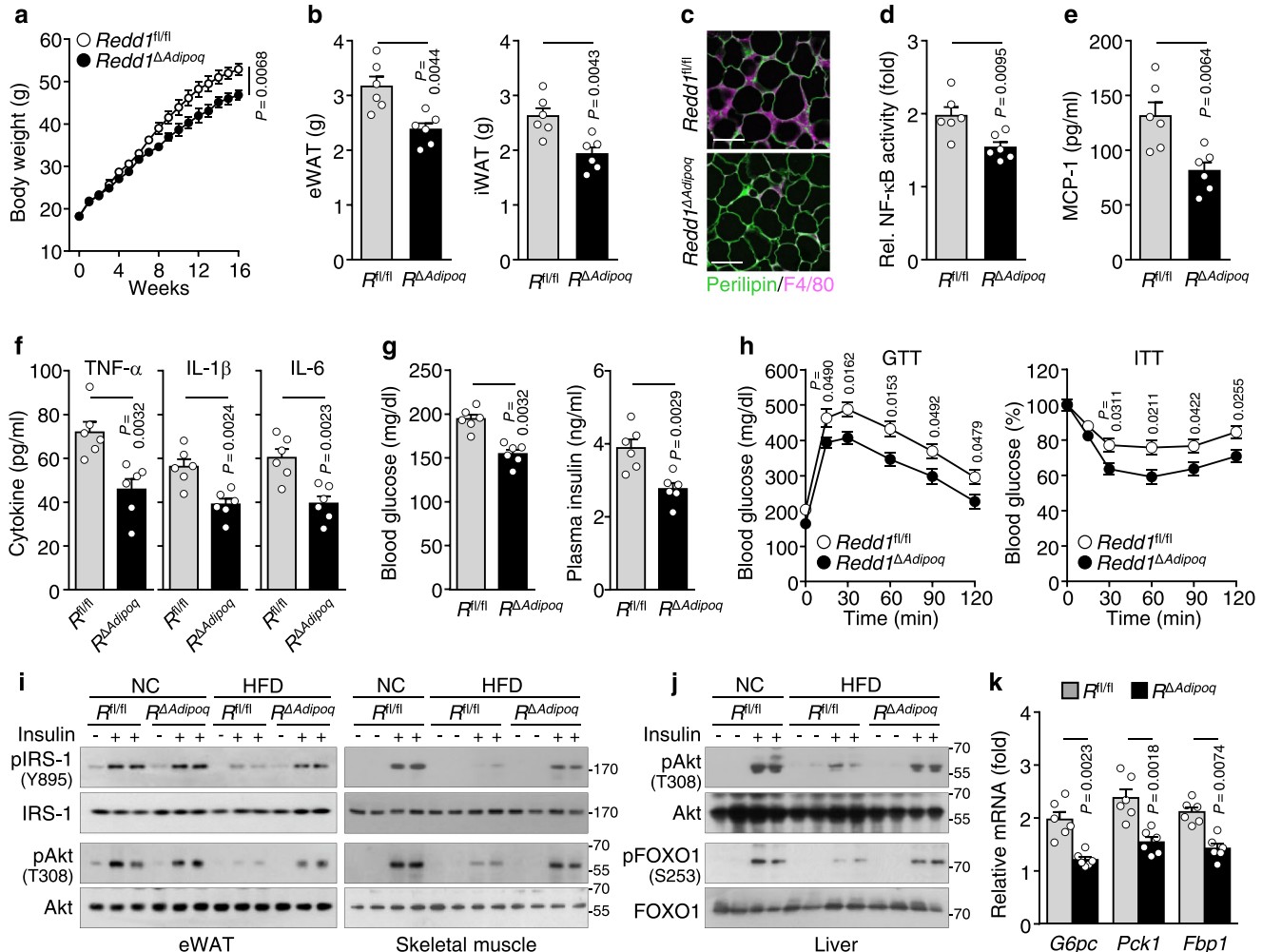

**Fig. 3 | Adipocyte *Redd1* deletion prevents HFD-induced obesity and inflammation. a** Weight gain over time in *Redd1*fl/fl (*R*fl/fl) and *Redd1*ΔAdipoq (*R*ΔAdipoq) mice fed HFD for 16 weeks (*n* = 6 per group). **b** Mass measurements for the eWAT and iWAT in HFD-fed *Redd1*fl/fl and *Redd1*ΔAdipoq mice (*n* = 6 per group). **c** Representative images showing perilipin (green) and F4/80 (purple) staining in the eWAT of *Redd1*fl/fl and *Redd1*ΔAdipoq mice fed HFD (*n* = 6 per group). Scale bar, 100 μm. **d** NF-κB activity in the eWAT from HFD-fed *Redd1*fl/fl and *Redd1*ΔAdipoq mice (*n* = 6 per group). **e, f** Plasma levels of inflammatory cytokines in HFD-fed *Redd1*fl/fl and *Redd1*ΔAdipoq mice (*n* = 6 per group). **g** Fasting plasma levels of glucose and insulin in HFD-fed *Redd1*fl/fl and

*Redd1*ΔAdipoq mice (*n* = 6 per group). **h** Assessment of GTT and ITT in HFD-fed *Redd1*fl/fl and *Redd1*ΔAdipoq mice after fasting for 12 and 6 h, respectively (*n* = 5 per group). **i, j** Representative western blots of phosphorylated IRS-1 and Akt in the eWAT and skeletal muscle (**i**) and phosphorylated Akt and FOXO1 in the liver (**j**) of NC- or HFD-fed *Redd1*fl/fl and *Redd1*ΔAdipoq mice after i.p. injection of saline or insulin (*n* = 3). **k** Relative expression levels of *G6pc*, *Pck1*, and *Fbp1* in the liver of HFD-fed *Redd1*fl/fl and *Redd1*ΔAdipoq mice compared with NC-fed mouse groups (*n* = 6 per group). Bar graphs represent mean ± s.e.m. Statistical significance was calculated using an unpaired two-tailed t-test. Source data are provided as a Source Data file.

muscle of *Redd1*fl/fl mice, both of which were restored in *Redd1*ΔLysM mice (Fig. 4i). Similarly, HFD intake resulted in a decrease in insulin-stimulated phosphorylation of Akt at Thr[308] and FOXO1 at Ser[253] in the liver of *Redd1*fl/fl mice, which was rescued in *Redd1*ΔLysM mice (Fig. 4j). Consequently, levels of *G6pc*, *Pck1*, and *Fbp1* expression were lower in the liver of HFD-fed *Redd1*ΔLysM than HFD-fed *Redd1*fl/fl mice, but not different between NC-fed groups (Fig. 4k and Supplementary Fig. 7d). These results suggest that in the macrophage, REDD1 is involved in insulin resistance and glucose metabolism dysregulation by stimulating meta-inflammation.

### REDD1 elicits adipogenesis and inflammation through atypical NF-κB activation

To investigate the functional role of REDD1 in adipogenesis, we compared the adipogenic potential of adipose SVF cells isolated from *Redd1*−/− mice and their WT littermates. When cultured in a differentiation medium containing an adipogenic cocktail (MDI) of methyl-isobutylxanthine, dexamethasone, and insulin, SVF cells from WT mice presented effective induction of REDD1 expression from 12 h

after the stimulation, followed by upregulation of PPARγ and C/EBPα expression on day 2 of the stimulation (Supplementary Fig. 8a), suggesting that REDD1 is upregulated early in the adipogenic differentiation process. As expected, MDI-stimulated *Redd1*-deficient SVF cells showed poorer adipogenesis than WT cells (Fig. 5a and Supplementary Fig. 8b). In addition, *Redd1* knockdown by shRNA suppressed adipogenic differentiation of 3T3-L1 preadipocytes cultured in the differentiation medium (Fig. 5b and Supplementary Fig. 8c). These results suggest that REDD1 is important for adipogenic differentiation. On the other hand, the inhibition of adipogenic differentiation and *Pparg* and *Cebpa* mRNA expression in *Redd1*ΔAdipoq mice SVF cells upon MDI stimulation was insignificant (Fig. 5c, d). This phenomenon is likely due to the deletion of *Redd1* only in matured *Redd1*ΔAdipoq adipocytes, as shown by the unchanged levels of REDD1 on day 4 after MDI stimulation and the marked downregulation on day 8 (Fig. 5e), consistent with previous studies that adiponectin expression was restricted in mature adipocytes[23]. However, lipogenic genes, such as acetyl-CoA carboxylase (*Acc*), fatty acid synthase (*Fasn*), and stearoyl-CoA desaturase-1 (*Scd-1*), were significantly downregulated in MDI-stimulated

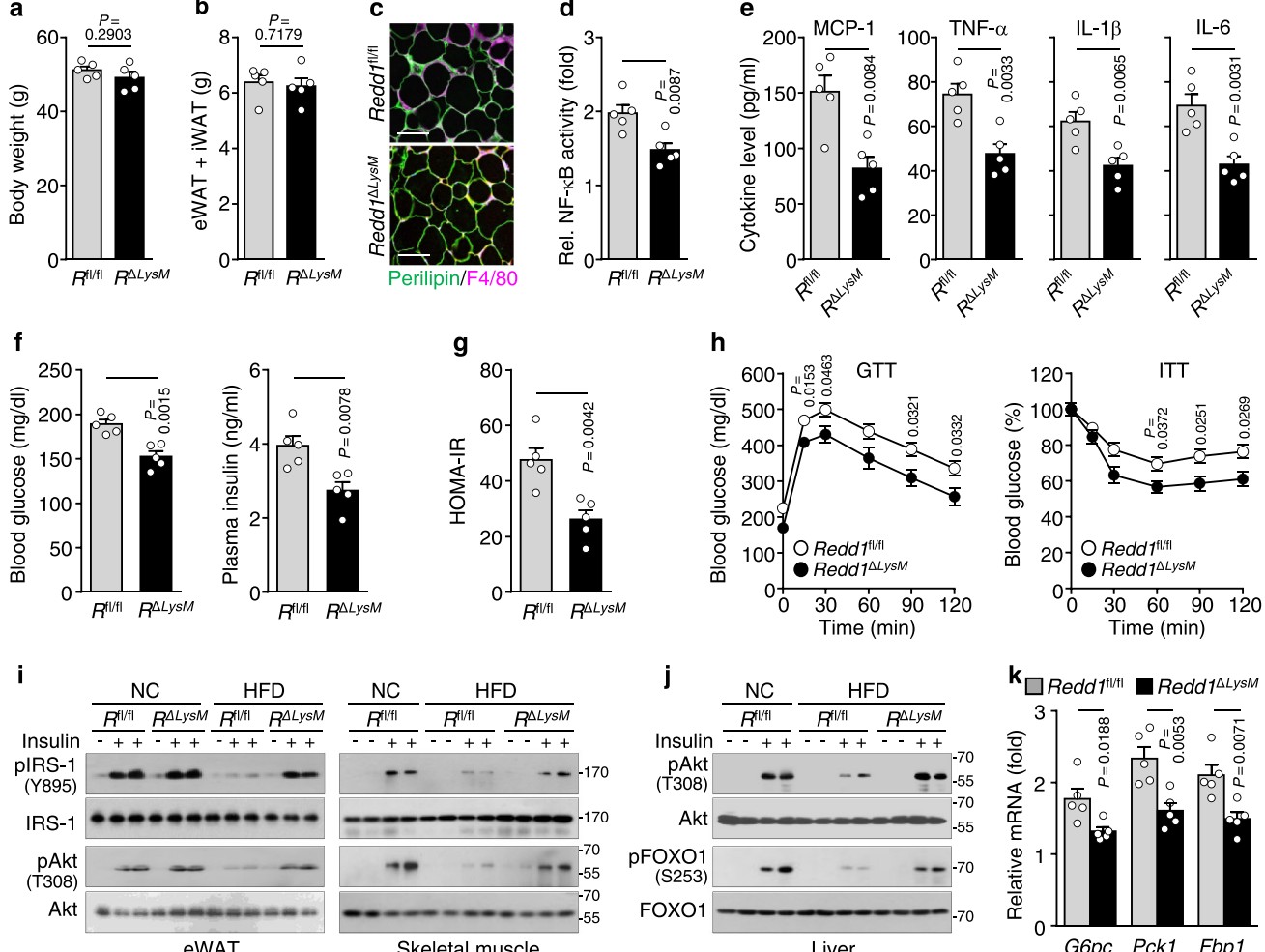

**Fig. 4 | Myeloid *Redd1* deficiency prevents HFD-induced meta-inflammation and metabolic dysregulation. a** Weight gain in *Redd1*fl/fl (*R*fl/fl) and *Redd1*ΔLysM (*R*ΔLysM) mice fed HFD for 16 weeks (*n* = 5 per group). **b** Measurement of fat (eWAT + iWAT) mass in HFD-fed *Redd1*fl/fl and *Redd1*ΔLysM mice (*n* = 5 per group). **c** Representative images showing perilipin (green) and F4/80 (purple) staining in the eWAT of HFD-fed *Redd1*fl/fl and *Redd1*ΔLysM mice (*n* = 5 per group). Scale bar, 100 μm. **d** NF-κB activity in the eWAT from HFD-fed *Redd1*fl/fl and *Redd1*ΔLysM mice (*n* = 5 per group). **e** Plasma levels of inflammatory cytokines in HFD-fed *Redd1*fl/fl and *Redd1*ΔLysM mice (*n* = 5 per group). **f** Fasting plasma levels of glucose and insulin in HFD-fed *Redd1*fl/fl and *Redd1*ΔLysM mice (*n* = 5 per group). **g** Calculation of HOMA-IR scores in HFD-fed

*Redd1*fl/fl and *Redd1*ΔLysM mice (*n* = 5 per group). **h** Assessment of GTT and ITT in HFD-fed *Redd1*fl/fl and *Redd1*ΔLysM mice fasting for 12 and 6 h, respectively (*n* = 5 per group). **i, j** Representative western blots of the insulin-responsive phosphorylation of IRS-1 and Akt in the eWAT and skeletal muscle (**i**) and phosphorylation of Akt and FOXO1 in the liver (**j**) of NC- or HFD-fed *Redd1*fl/fl and *Redd1*ΔLysM mice (*n* = 3). **k** Relative expression levels of *G6pc*, *Pck1*, and *Fbp1* in the liver of HFD-fed *Redd1*fl/fl and *Redd1*ΔLysM mice compared with NC-fed mouse groups (*n* = 5 per group). Bar graphs represent mean ± s.e.m. Statistical significance was calculated using an unpaired two-tailed *t*-test. Source data are provided as a Source Data file.

*Redd1*ΔAdipoq SVF cells (Fig. 5f). This suggests that REDD1 does not affect the differentiation of SVF cells into adipocytes in *Redd1*ΔAdipoq mice fed HFD but can inhibit fatty acid synthesis.

Next, we investigated the role of NF-κB in REDD1-dependent adipogenesis, because REDD1 activates NF-κB[18], which plays an important role in adipogenic differentiation[24,25]. Adenoviral overexpression of REDD1 in 3T3-L1 cells resulted in an increase in NF-κB-driven luciferase activity, which was suppressed by overexpressing IκBα or knocking down NF-κB p65, but not IKKα or IKKβ (Fig. 5g and Supplementary Fig. 8d), consistent with a previous study showing that REDD1 activates the atypical NF-κB pathway by sequestering IκBα[18]. Furthermore, REDD1 overexpression promoted adipogenesis and *Pparg* and *Cebpa* expression in 3T3-L1 cells, which was attenuated upon overexpression of IκBα (Fig. 5h, i and Supplementary Fig. 8e). These results suggest that REDD1 stimulates adipogenic differentiation through NF-κB activation, consistent with other results[11,24,25]. We next examined whether NF-κB activation can be stimulated during adipocyte differentiation. When cultured in the differentiation medium, adipose SVF cells and

3T3-L1 cells showed increased REDD1 expression, NF-κB p65 nuclear translocation, and NF-κB-reporter activity without affecting IκBα levels, as shown by REDD1 overexpression (Supplementary Fig. 8f, g). This suggests that the REDD1–NF-κB axis plays an important role in adipocyte differentiation, consistent with previous studies showing that NF-κB plays an important role in adipogenic differentiation[25,26].

Since the adipogenic genes, *Pparg* and *Cebpa*, are known to be upregulated by degradation of β-catenin through IKKβ-mediated β-catenin phosphorylation or NF-κB-induced Smurf2 expression[11,24,27], we examined the role of REDD1 in the expression of these genes. REDD1 overexpression increased PPARγ and CEBPα levels without affecting IKKαβ phosphorylation or nuclear β-catenin accumulation in 3T3-L1 cells (Supplementary Fig. 8h), indicating that REDD1 promotes adipogenesis independent of β-catenin degradation. Consistent with the previous study showing NF-κB-dependent transcription of CEBPα[28], six putative NF-κB binding sites were predicted on *Cebpa* promoter using the ALGGEN PROMO software v8.3. (http://alggen.lsi.upc.es) (Supplementary Fig. 8i). Among them, the proximal site centered at −1052 bp

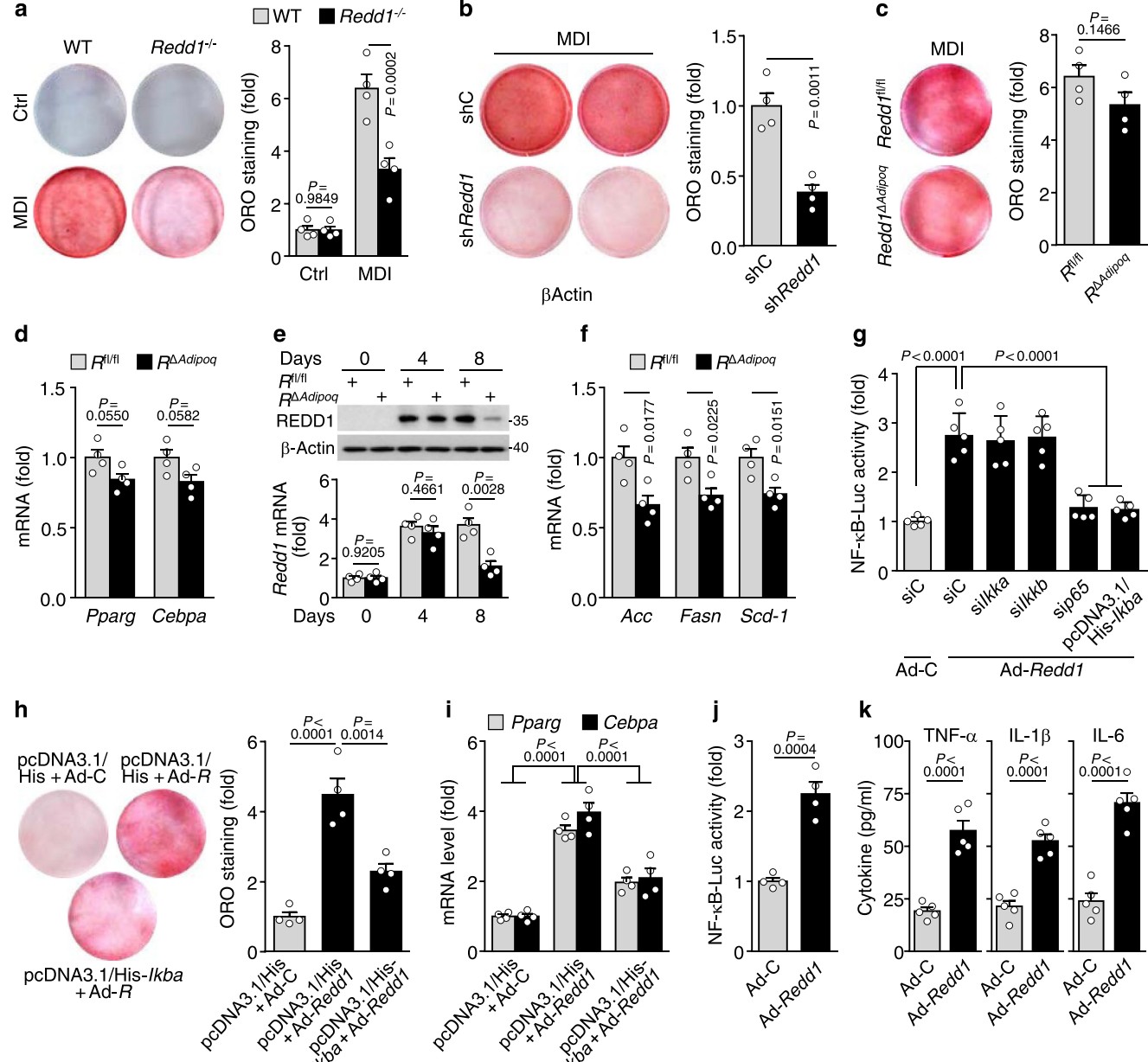

**Fig. 5 | REDD1 elicits adipocyte differentiation and macrophage inflammation through NF-κB activation. a–c** Representative oil red-O (ORO)-stained images of WT and *Redd1⁻/⁻* SVF cells (**a**), shControl (shC)- or sh-Redd1-transfected 3T3-L1 cells (**b**), and WT (*Redd1ᶠˡ/ᶠˡ*, *Rᶠˡ/ᶠˡ*) and *Redd1ᐃAdipoq* (*RᐃAdipoq*) SVF cells (**c**) when cultured in differentiation medium (MDI) and quantification of relative ORO intensity (*n* = 4). **d–f**, Expression levels of adipogenic genes (**d**), REDD1 (**e**), and lipogenic genes (**f**) in *Rᶠˡ/ᶠˡ* and *RᐃAdipoq* SVF cells cultured in MDI medium and quantification of relative ORO intensity (*n* = 4). **g** Assessment of NF-κB–Luc activity in 3T3-L1 cells transfected either with siRNA for control, *Ikka*, *Ikkb*, or NF-κB *p65* (*p65*) or with pcDNA3.1/His-*Ikba* (*n* = 5). **h, i** Representative images and realative quantification of ORO-stained images (**h**) and expression levels of *Pparg* and *Cebpa* (**i**) in 3T3-L1 cells infected with control adenovirus (Ad-C) or adenoviral *Redd1* (Ad-*R*) after transfection with vector alone or pcDNA3.1/His-*Ikba* (*n* = 4). **j** NF-κB–Luc activity in mouse peritoneal macrophages infected with Ad-C or Ad-*Redd1* (*n* = 4). **k** Cytokine production in mouse peritoneal macrophages infected with Ad-C or Ad-*Redd1* (*n* = 5). Bar graphs represent mean ± s.e.m. Statistical significance was calculated using one-way ANOVA (**g, h**) and two-way ANOVA (**a, i**) followed by the Holm–Sidak post hoc test and an unpaired two-tailed *t*-test (**b–f, j, k**). Source data are provided as a Source Data file.

had higher transcription activity than others, which was confirmed in *Redd1*-overexpressing cells using chromatin immunoprecipitation (Chip) assay and promoter activity analysis (Supplementary Fig. 8j, k). This suggests that REDD1-induced NF-κB activation increases pre-adipocyte differentiation by transcriptional upregulation of *Cebpa* and in turn the positive cross-regulation loop between *Cebpa* and *Pparg* expression[29]. However, further detailed function of each site needs to be analyzed. In addition, REDD1 overexpression increased NF-κB-driven luciferase activity as well as TNF-α, IL-1β, and IL-6 production in mouse peritoneal macrophages (Fig. 5j, k and Supplementary Fig. 8l),

which is consistent with the previous results demonstrating the proinflammatory action of REDD1 in RAW264.7 cells through atypical NF-κB activation[18]. Thus, REDD1 stimulates adipogenesis and proinflammatory cytokine production through the atypical activation of NF-κB by binding to and sequestering IκBα[18].

## Lys²¹⁹/²²⁰ of REDD1 are crucial for NF-κB activation, adipogenesis, and inflammation

The C-terminal region of REDD1, including the amino acid sequence spanning residues 178–229, plays a key role in atypical NF-κB activation

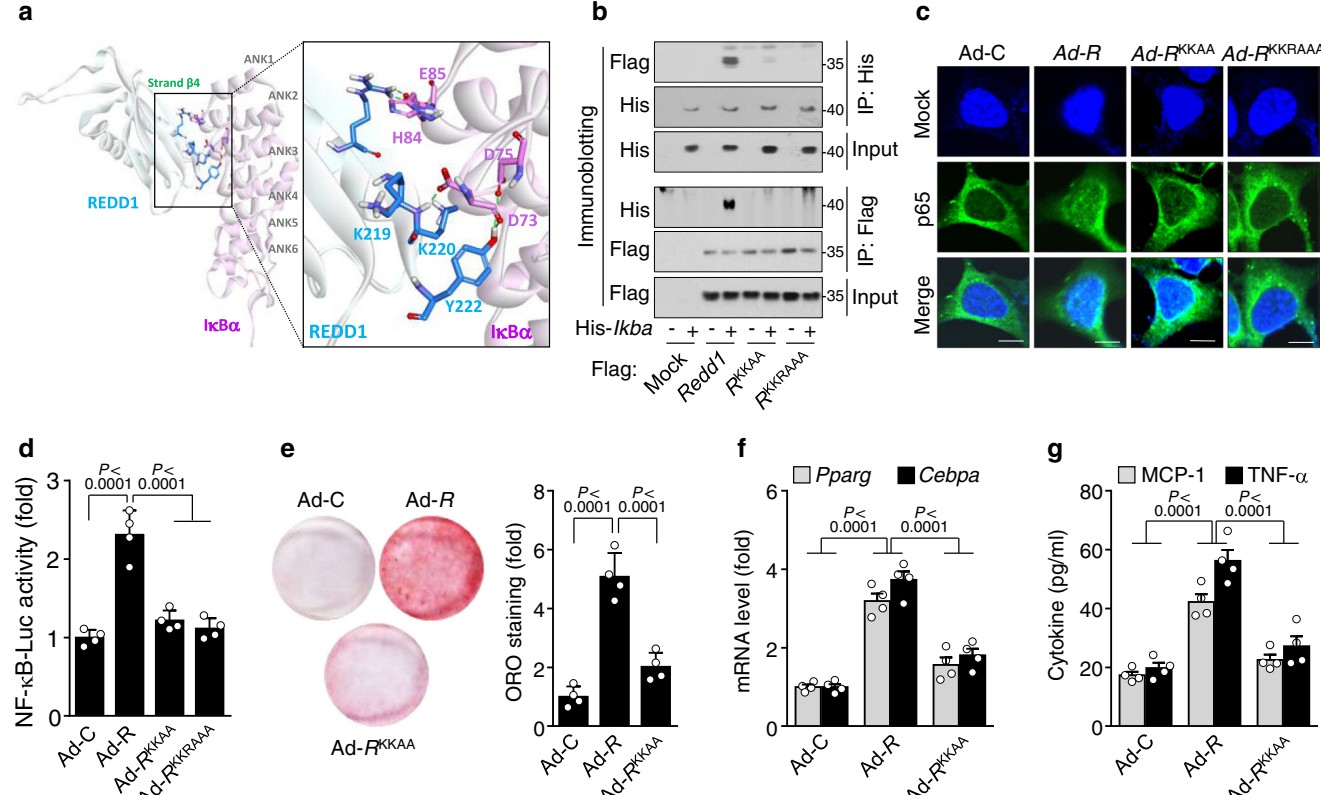

**Fig. 6 | Lys²¹⁹/²²⁰ of REDD1 are crucial for NF-κB activation, adipogenesis, and inflammation. a** Predictive binding conformation between REDD1 and IκBα using computational protein-protein molecular docking methods. **b** Co-immunoprecipitation analysis of the interaction between REDD1 and IκBα in HEK293 cells transfected with pcDNA3.1/His-*Ikba* (His-*Ikba*) and either pFlag-CMV-1-*Redd1* (*Redd1*) or *Redd1* mutants (*R*^KKAA^ and *R*^KKRAAA^) (*n* = 3). **c** Representative confocal images of NF-κB p65 nuclear translocation in HEK293 cells infected with Ad-C, Ad-*Redd1*, or its mutants (*n* = 4). Scale bar, 50 μm. **d** Assessment of NF-κB-Luc activity in 3T3-L1 cells infected with Ad-control, Ad-*Redd1*, or its mutants (*n* = 4).

**e** Representative ORO-stained images of 3T3-L1 cells infected with Ad-control, Ad-*Redd1*, or Ad-*Redd1*^KKAA^ and quantification of relative ORO intensity (*n* = 4). **f** Expression levels of *Pparg* and *Cebpa* in 3T3-L1 cells infected with Ad-control, Ad-*Redd1*, or Ad-*Redd1*^KKAA^ (*n* = 4). **g** Production of MCP-1 and TNF-α in macrophages infected with Ad-control, Ad-*Redd1*, or Ad-*Redd1*^KKAA^ (*n* = 4). Bar graphs represent mean ± s.e.m. Statistical significance was calculated using one-way ANOVA (**d**, **e**) and two-way ANOVA (**f**, **g**) followed by the Holm−Sidak post hoc test. Source data are provided as a Source Data file.

by binding to and sequestering IκBα[18]. We predicted a reasonable binding conformation between REDD1 and IκBα using computer modeling methods. Several well-known protein-protein docking servers, including HADDOCK and HDOCK, were employed for the study. Both results revealed a similar binding shape of the entire protein structure and the interaction networks of the key residues; however, the binding mode obtained from HADDOCK was more compatible with the previously established experimental data[18], when compared with the interactions between IκBα and NF-κB p65[30] (Supplementary Fig. 9a−d). The proposed model revealed that Lys²¹⁹, Lys²²⁰, and Tyr²²², located at the functional hotspot on strand β4 (residues 214−220) of REDD1[31], are likely to form hydrogen bonds with Asp⁷³ and Asp⁷⁵ in ANK1 (residues 67−130) of IκBα (Fig. 6a and Supplementary Fig. 9c). In addition, Arg¹³³ in helix α2 of REDD1 contributes to potential interactions with His⁸⁴ and Glu⁸⁵ in ANK1 of IκBα. These interactions may interfere with the binding of p65 to IκBα by hindering the approach of Lys³⁰¹, Arg³⁰², and Arg³⁰⁴ on the nuclear localization signal (NLS, ³⁰¹Lys-Arg-Lys-Arg³⁰⁴) of p65[30]. These data indicate that the sequestration of IκBα by REDD1 allows the nuclear import of NF-κB by unmasking the NLS of p65. We next examined the amino acids Asp⁷³, Asp⁷⁵, and Arg¹³³ in REDD1, which are essential for interaction with IκBα and activation of NF-κB by performing substitution mutations. Overexpression of mutant REDD1 with Lys²¹⁹Ala/Lys²²⁰Ala (*Redd1*^KKAA^) or Lys²¹⁹Ala/Lys²²⁰Ala/Arg¹³³Ala (*Redd1*^KKRAAA^) effectively and comparably inhibited REDD1 binding with IκBα (Fig. 6b), as well as prevented the nuclear translocation of NF-κB p65 and NF-κB-driven luciferase activity

without affecting IKKα/β phosphorylation and IκBα degradation (Fig. 6c, d and Supplementary Fig. 9e). Overexpressed *Redd1*^KKAA^ could not stimulate adipogenic differentiation and *Pparg* and *Cebpa* expression in 3T3-L1 cells (Fig. 6e, f) and the production of MCP-1 and TNF-α in macrophages (Fig. 6g). These results suggest that both lysine residues of REDD1 are essential for inducing adipogenic differentiation and cytokine production through the atypical activation of NF-κB by binding to and sequestration of IκBα.

### *Redd1*^KKAA^ mice lack the HFD-induced obesity phenotype
Using *Redd1*^KKAA^ mice with no effect on the expression of mutant REDD1 (Supplementary Fig. 1e), we investigated whether REDD1-mediated NF-κB activation is crucial for HFD-induced metabolic dysfunction and inflammation. Under HFD conditions, *Redd1*^KKAA^ mice had lower body weight, eWAT and iWAT mass, and *Pparg* and *Cebpa* expression in the eWAT than their WT littermates; however, these phenotypic characteristics were similar in both groups when fed NC (Fig. 7a−c and Supplementary Fig. 10a−c). The protective effects of *Redd1*^KKAA^ on HFD-induced adipogenesis were also confirmed by the significant restoration of the plasma levels of leptin, resistin, and adiponectin (Supplementary Fig. 10d−f). Compared with WT mice, *Redd1*^KKAA^ mice showed a less HFD-mediated increase in adipocyte size, F4/80⁺ macrophage population, and CLS formation in the eWAT; however, these characteristic differences were not observed between NC-fed groups (Fig. 7d and Supplementary Fig. 10g, h). Consistently, *Redd1*^KKAA^ mice had reduced the HFD-induced increase in NF-κB activity

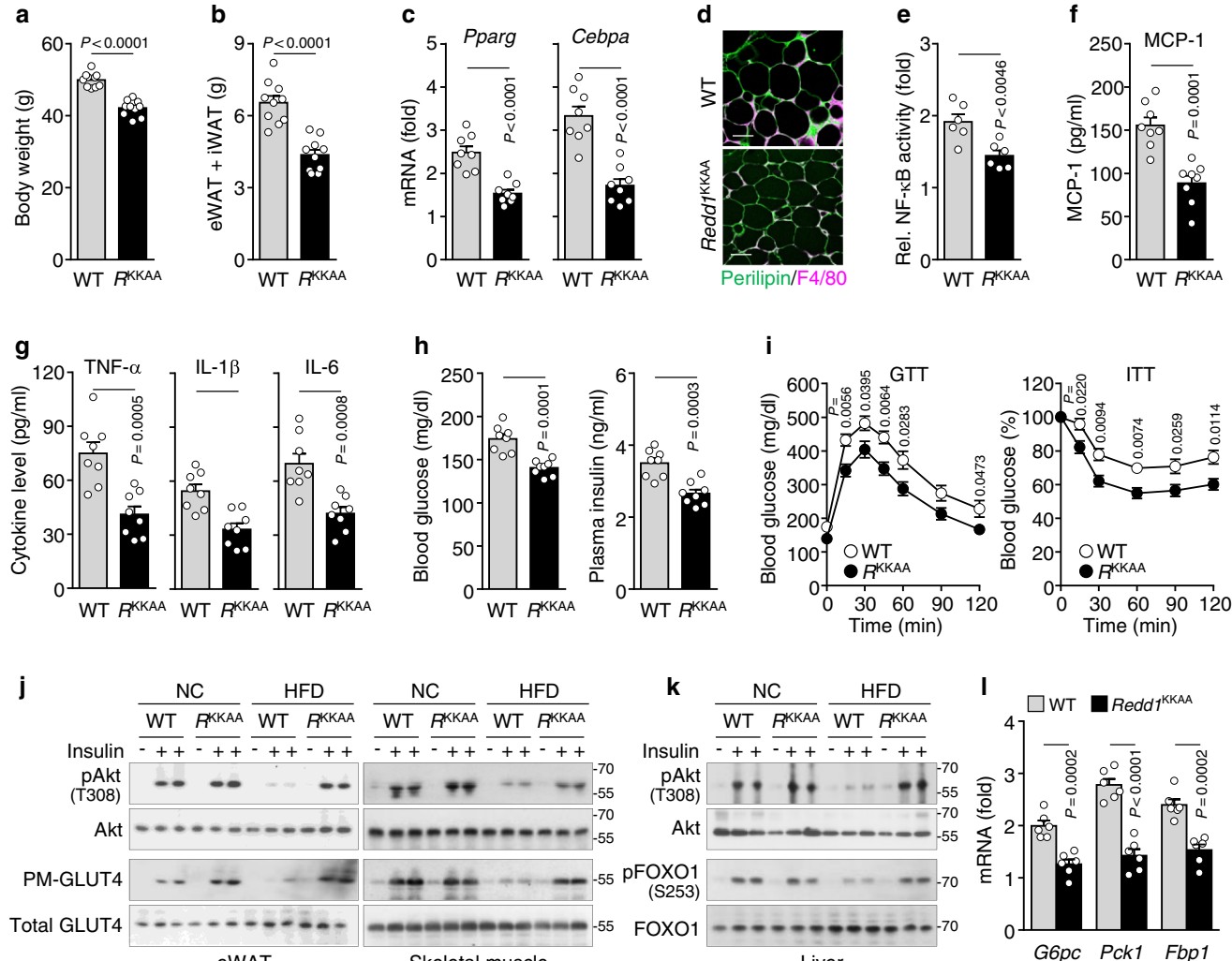

**Fig. 7 | HDF-induced obesity and metabolic phenotypes are prevented in *Redd1*^KKAA mice. a** Weight gain in WT and *Redd1*^KKAA mice after being fed HFD for 16 weeks (*n* = 10 per group). **b** eWAT and iWAT mass measurements in HFD-fed *Redd1*^KKAA mice and their WT littermates (*n* = 10 per group). **c** Expression levels of *Pparg* and *Cebpa* in the eWAT of *Redd1*^KKAA mice and WT littermates fed HFD for 10 weeks (*n* = 8 per group). **d** Representative images showing perilipin (green) and F4/80 (purple) staining in the eWAT of HFD-fed *Redd1*^KKAA mice and WT littermates (*n* = 5 per group). Scale bar, 100 μm. **e** NF-κB activity in the eWAT from HFD-fed *Redd1*^KKAA mice and WT littermates (*n* = 6 per group). **f, g** Plasma levels of inflammatory cytokines in HFD-fed *Redd1*^KKAA mice and WT littermates (*n* = 8 per group). **h** Fasting plasma levels of glucose and insulin in HFD-fed *Redd1*^KKAA mice and WT littermates (*n* = 8 per group). **i** Assessment of GTT and ITT in HFD-fed *Redd1*^KKAA mice and WT littermates after fasting for 12 and 6 h, respectively (*n* = 6 per group). **j, k** Representative western blots of insulin-responsive Akt phosphorylation and plasma membrane-associated GLUT4 (PM-GLUT4) levels in the eWAT and skeletal muscle (**j**) and Akt and FOXO1 phosphorylation in the liver (**k**) of HFD-fed *Redd1*^KKAA mice and WT littermates (*n* = 3). **l** Relative expression levels of *G6pc*, *Pck1*, and *Fbp1* in the liver of HFD-fed *Redd1*^KKAA mice and WT littermates compared with NC-fed mice (*n* = 6 per group). Bar graphs represent mean ± s.e.m. Statistical significance was calculated using an unpaired two-tailed *t*-test. Source data are provided as a Source Data file.

without affecting IκBα levels in eWAT and MCP-1, TNF-α, IL-1β, and IL-6 protein and mRNA levels in serum and eWAT, respectively, when compared with those in WT mice (Fig. 7e–g and Supplementary Fig. 10i–l). These results suggest that Lys^219/220 are crucial for REDD1-induced adipogenesis and meta-inflammation. Next, we determined the levels of blood glucose and insulin in *Redd1*^KKAA mice. When fed HFD, *Redd1*^KKAA mic exhibited significantly lower fasting glucose and insulin levels than WT littermates; however, the levels were not different between NC-fed groups (Fig. 7h and Supplementary Fig. 10m), suggesting that *Redd1*^KKAA mice do not present the HFD-induced diabetic phenotype. GTT and ITT analyses revealed that *Redd1*^KKAA mice had improved glucose tolerance and insulin sensitivity compared with their WT littermates when fed only HFD (Fig. 7i and Supplementary Fig. 10n). Moreover, knock-in of *Redd1*^KKAA mutations rescued HFD-induced inhibition of insulin-stimulated Akt phosphorylation and GLUT4 translocation to the plasma membrane in the eWAT and

skeletal muscle of WT mice (Fig. 7j). In addition, the knock-in mutations also restored HFD-induced suppression of insulin-mediated phosphorylation of Akt and FOXO1 in the liver of WT mice (Fig. 7k), resulting in the downregulated expression of the hepatic gluconeogenic genes *G6pc*, *Pck1*, and *Fbp1* in HFD-fed *Redd1*^KKAA mice (Fig. 7l and Supplementary Fig. 10o). Thus, both lysine residues of REDD1 are functionally involved in the impairment of insulin signaling and glucose metabolism by inducing atypical NF-κB activation and meta-inflammation.

**Global or adipocyte-specific loss of *Redd1* prevents HFD-induced hepatic steatosis**

Since obesity is associated with fatty liver, we examined the role of REDD1 in HFD-induced hepatic steatosis. HFD feeding resulted in effectively developed hepatic steatosis in WT or *Redd1*^fl/fl mice, as evidenced by hematoxylin and eosin (H&E) staining of the liver tissues,

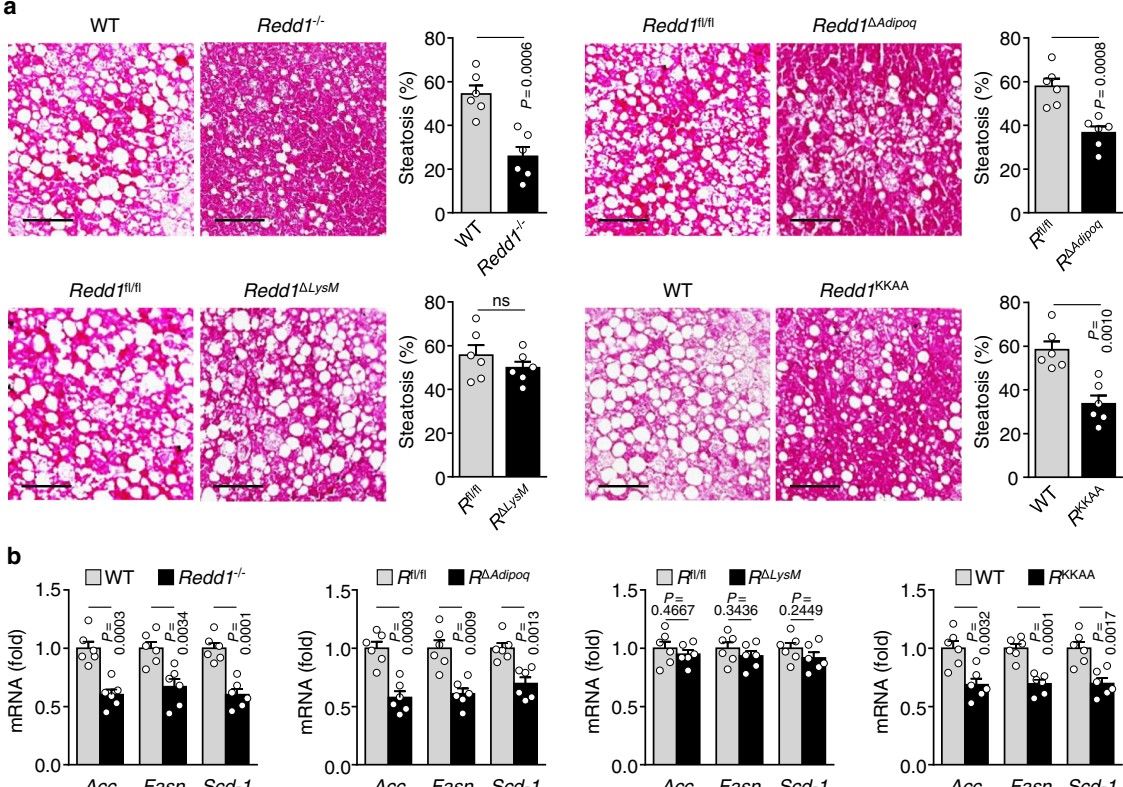

**Fig. 8 | Global or adipocyte-specific loss of *Redd1* prevents HFD-induced hepatic steatosis. a** Representative images of H&E-stained liver tissues from HFD-fed *Redd1*[−/−], *Redd1*[ΔAdipoq], *Redd1*[ΔLysM], *Redd1*[KKAA], and control mice, and quantification of hepatic steatosis from H&E-stained liver tissues (*n* = 6 per group). Scale bars, 100 μm. **b** Expression levels of *Acc*, *Fasn*, and *Scd-1* in the liver of HFD-fed *Redd1*[−/−], *Redd1*[ΔAdipoq], *Redd1*[ΔLysM], *Redd1*[KKAA], and their control mice (*n* = 6 per group). Bar graphs represent mean ± s.e.m. Statistical significance was calculated using an unpaired two-tailed *t*-test. Source data are provided as a Source Data file.

and the characteristic features of steatosis were significantly prevented in *Redd1*[−/−], *Redd1*[ΔAdipoq], and *Redd1*[KKAA] mice, consistent with a recent study[16], but not in *Redd1*[ΔLysM] mice; however, no hepatic steatosis was observed in NC-fed mice (Fig. 8a and Supplementary Fig. 11a). The histological features of steatosis are highly correlated with circulating levels of alanine aminotransaminase (ALT), an index of hepatic injury (Supplementary Fig. 11b). In addition, the expression of hepatic lipogenic genes, including *Acc*, *Fasn*, and *Scd-1*, were decreased in *Redd1*[−/−], *Redd1*[ΔAdipoq], and *Redd1*[KKAA] mice, but not in *Redd1*[ΔLysM] mice on HFD (Fig. 8b). This indicates that REDD1 regulates lipogenesis and lipid homeostasis by the interplay between metabolically active organs such as liver and adipose tissue, but not immune cells. Thus, our findings suggest that ablation of *Redd1* in the whole body or adipocytes, but not in myeloid cells, prevents HFD-induced hepatic steatosis by suppressing atypical NF-κB-dependent lipogenesis rather than meta-inflammation.

## Discussion

Several studies have demonstrated that REDD1 expression is upregulated in various pathophysiological conditions, including obesity and T2D[15,16,20]. However, the functional role of REDD1 has not been distinctly identified in obesity-induced meta-inflammation and metabolic dysregulation. We used cell-type-specific *Redd1*-deficient mice to reveal that HFD-induced REDD1 regulates adipogenesis, meta-inflammation, the diabetic phenotype, and hepatic steatosis in an adipocyte- or macrophage-dependent manner. Loss- and gain-of-function studies also demonstrated that REDD1 stimulates adipocyte differentiation and macrophage-mediated cytokine production via atypical NF-κB activation by binding to and sequestering IκBα. Together with other results[18,19], these data indicate that the REDD1-NF-κB signaling axis may

be different from the canonical pathway associated with IKKβ-dependent IκBα degradation in acute immune responses. Thus, REDD1 may be a crucial link between obesity, meta-inflammation, and T2D through the atypical activation of NF-κB.

REDD1 is an endogenous inhibitor of mTORC1 that promotes protein and lipid biosynthesis and causes metabolic disorders[32–34]. Genetic or pharmacological inhibition of the mTORC1 pathway prevents HFD-induced adipogenesis and obesity[34–37]. Thus, REDD1 may negatively regulate adipogenesis and metabolic homeostasis by inhibiting the mTORC1 pathway. Unexpectedly, our data revealed that *Redd1*[−/−], *Redd1*[ΔAdipoq], and *Redd1*[KKAA] mice do not present HFD-induced obesity and metabolic dysregulation, which is consistent with other results[38–40]. Notably, HFD increased phosphorylation of ribosomal protein S6 kinase (S6K) and ribosomal protein S6, the downstream targets of mTORC1, in the liver, skeletal muscle, and eWAT of *Redd1*[−/−], *Redd1*[ΔAdipoq], *Redd1*[KKAA], and control mice; however, the phosphorylation levels were not different between *Redd1*-targeted mice and control littermates under the same feeding conditions (Supplementary Fig. 12a–c). These observations were consistent with previous results showing a similar increase of mTORC1 activity in skeletal muscle of both HFD-fed *Redd1*[−/−] and WT mice[20] but different from other results[16,41,42]. For instance, mTORC1 activity was unexpectedly decreased in the liver of HFD-fed *Redd1*[−/−] mice[16] and unchanged in *Redd1*-disrupted hepatocyte and *Redd1*-overexpressing C3H10T1/2 adipocytes[41,42]. Interestingly, we found that REDD1 overexpression inhibited the mTORC1 signaling pathway in cultured endothelial cells[43] and 3T3-L1 preadipocytes (Supplementary Fig. 12d). Although the mechanisms involved remain unknown, our findings suggest that REDD1 promotes rather than prevents obesity-induced metabolic disorders in an mTORC1-independent manner.

There are a few studies describing the role of REDD1 in metabolic dysregulation and hepatic steatosis in HFD-induced obese mouse models[16,20]. Williamson et al. reported that *Redd1*−/− mice showed reduced weight gain and lower blood glucose levels than WT mice under HFD conditions[20], which is consistent with our findings. On the other hand, Dumas et al. demonstrated that in HFD-fed mice, *Redd1* deficiency prevented hepatic steatosis by decreasing the expression of lipogenic enzymes including SREBP-1c, FASN, and SCD-1, without affecting weight gain and glucose intolerance[16]. These results are in part different from our findings that global or adipocyte-specific *Redd1* deletion protected mice from HFD-induced obesity and hepatic steatosis. In general, chronic lipid overload, such as HFD feeding, elicits lipid redistribution between adipose tissue and liver, resulting in a similar metabolic phenotype in both organs through lipid homeostasis; however, the contrasting effects (protective or non-protective) of *Redd1* deficiency on metabolic dysregulation i.e., obesity in an HFD-fed mouse model could arise from complex interactions among genetic background, nutrient composition, and environmental stress[44–47].

REDD1 is a cellular inhibitor of NF-κB[18,19], suggesting that it can regulate adipogenesis and obesity through the activation of NF-κB. Consistently, a study by Helsley et al. demonstrated that genetic and pharmacological inhibition of the NF-κB pathway attenuates differentiation of 3T3-L1 preadipocytes and human adipose stem cells into adipocytes in vitro[11]. They also showed that adipocyte-specific *Ikkb* deletion prevents adipocyte differentiation, adipogenesis, and weight gain in HFD-fed mice. In addition, mice lacking either NF-κB *p50* or *Ikke* are protected from HFD-induced adipogenesis and obesity, probably because they increase their catabolic rate and energy expenditure[25,48]. Moreover, WT mice, but not *Redd1*-deficient mice, showed increased NF-κB activation in multiple organs, including adipose tissue, after exposure to endotoxin or cigarette smoke[18,19,49], indicating that REDD1 can activate NF-κB signaling. These findings suggest that HFD-induced REDD1 expression may contribute to adipocyte differentiation and obesity through NF-κB activation. Therefore, ablation or inhibition of the NF-κB pathway significantly decreases the expression of adipogenic genes, including *Pparg* and *Cebpa*, in adipose SVF cells[11,24], indicating that NF-κB directly promotes adipogenesis. Consistently, we found that *Redd1*−/− and *Redd1*ΔAdipoq mice, but not *Redd1*ΔLysM mice, were resistant to HFD-induced adipogenesis and obesity. In addition, REDD1 overexpression stimulated *Cebpa* expression in an NF-κB-dependent manner, leading to the promotion of adipogenic differentiation in 3T3-L1 preadipocytes through cross-transcriptional upregulation of *Cebpa* and *Pparg*[29], which were suppressed by the overexpression of IκBα. These findings suggest that REDD1 plays a crucial role in adipogenesis in an NF-κB-dependent manner.

Notably, we found that the development of HFD-induced obesity was prevented in *Redd1*ΔAdipoq mice; however, their SVF cells did not affect adipogenic gene expression and adipogenesis in vitro but inhibited lipogenic gene expression. This suggests that *Redd1* deletion in mature adipocytes of *Redd1*ΔAdipoq mice, as previously reported[23], does not affect adipocyte differentiation (including mitotic clonal expansion) but can inhibit fatty acid synthesis (adipocyte hypertrophy). Adipocyte hypertrophy or lipohypertrophy, which is essential for adipose tissue growth and weight gain, results from not only de novo fatty acid synthesis but also impaired lipolysis or energy expenditure. The reduced WAT mass and body weight in *Redd1*ΔAdipoq mice may be associated with decreased fatty acid synthesis or increased lipolysis or energy expenditure, consistent with previous studies that genetic inhibition of the NF-κB pathway increased fatty acid catabolism and energy expenditure in a mouse model of HFD-induced obesity[25,48]. Therefore, the function of the preadipocyte-specific REDD1/NF-κB pathway in adipogenesis, lipid metabolism, and energy expenditure, should be investigated in more detail in a mouse model of diet-induced obesity using adipocyte progenitor cell-specific *Redd1*-deleted mice, such as *Redd1*ΔPdgfRa mice, as previously reported[23].

Meta-inflammation has been implicated as a critical molecular link between obesity and metabolic disorders, including insulin resistance and T2D. Emerging evidence has revealed that several obesity-related factors upregulate REDD1[20,50–52], which stimulates NF-κB activation and proinflammatory cytokine production[18,19,49]. Proinflammatory cytokines act as crucial factors in metabolic syndromes by disrupting insulin signaling[3]. We and other researchers have shown that *Redd1* deficiency prevents NF-κB-dependent proinflammatory cytokine production in the lungs and adipose tissues of mice exposed to endotoxins[18,19,53]. REDD1 overexpression is sufficient for NF-κB activation and induction of cytokine gene expression through a mechanism in which REDD1 directly interacts with IκBα, without altering IKKβ phosphorylation and IκBα degradation[18], suggesting that REDD1 stimulates atypical NF-κB activation by interactively sequestering IκBα rather than degrading it. Therefore, REDD1-mediated NF-κB activation was blocked by IκBα overexpression, but not by the knockdown of IKKα and IKKβ. In fact, both Lys219 and Lys220 of REDD1 may form hydrogen bonds with Asp73 of IκBα to mask the NLS of NF-κB p65 and prevent the nuclear translocation of NF-κB. Thus, overexpression of REDD1KKAA did not induce NF-κB activation, preadipocyte differentiation, and cytokine production in vitro; similar to *Redd1*−/− mice, *Redd1*KKAA mice presented restrained obesity-induced characteristics, including proinflammatory cytokine production and insulin resistance. These findings suggest that REDD1-driven atypical NF-κB activation is crucial for meta-inflammation and metabolic dysregulation. Notably, the REDD1−NF-κB axis may be different from the IKKαβ-dependent canonical and non-canonical NF-κB pathways associated with innate and adaptive immune systems[54].

The prolonged or persistent expression of REDD1 under obese conditions maintains low and moderate levels of NF-κB activation, resulting in the induction of meta-inflammation. These unique properties suggest that REDD1 plays a crucial role in the initiation of meta-inflammation, which is a link between obesity and metabolic disorders. Consistently, in *Redd1*ΔLysM mice, HFD-induced inflammatory cytokine production was suppressed, without inhibiting adipogenesis, and thus, insulin sensitivity and glucose metabolism were improved. However, *Redd1*ΔAdipoq mice were resistant to both adipogenesis and cytokine production, suggesting that REDD1 promotes adipogenesis and meta-inflammation in adipocytes through NF-κB activation. MCP-1 production in adipocytes induced by REDD1 operates a positive feedback loop for the amplification of macrophage infiltration and proinflammatory cytokine production in adipose tissues, which leads to insulin resistance. These findings suggest that REDD1 causes meta-inflammation and insulin resistance in obese mice through the activation of NF-κB, although REDD1 performs some cell-type-specific functions in adipocytes and macrophages.

Proinflammatory cytokines are elevated in adipose tissue or serum samples derived from rodents or humans with obesity[4,55,56], suggesting that they serve as a pathogenic link between obesity, insulin resistance, and T2D[3]. This concept was proven in cytokine-deficient mice and by the administration of exogenous cytokines or targeting antibodies in obese animals and patients with T2D[4,57–60]. The cytokines, such as TNF-α and IL-1β, stimulate the inhibitory phosphorylation of multiple serine/threonine residues on IRS-1 and IRS-2 by activating JNK and IKKβ through their receptors, thereby preventing insulin-stimulated tyrosine phosphorylation of IRS-1 and IRS-2 and their ability to bind PI3K, and thus, inhibiting the phosphorylation of its downstream effector Akt[61–63]. In addition, IL-6 plays an important role in negatively regulating insulin signaling through inhibition of IRS-1 and IRS-2 phosphorylation and/or promotion of ubiquitin-mediated IRS-1 and IRS-2 degradation by stimulating Stat3-dependent SOCS expression[21,64]. Indeed, *Redd1*−/− mice had reduced HFD-induced IL-6

production, Stat3 phosphorylation, and *Socs3* expression. These results suggest that HFD-induced REDD1 elicits meta-inflammation and subsequently impairs insulin-driven IRS–PI3K–Akt signaling. As a result, *Redd1*-deficient mice showed improved GLUT4 translocation into the plasma membrane of the skeletal muscle and increased phosphorylation-dependent inactivation of FOXO1 in the liver, along with the downregulation of gluconeogenic gene expression, resulting in improved insulin resistance and glucose metabolism. Thus, our findings suggest that REDD1-mediated atypical NF-κB activation and proinflammatory cytokine production in lipid-laden adipocytes and adipose tissue macrophages are causative factors for inducing global insulin resistance, consistent with results reported in NF-κB *p50*−/− mice[25], further underscoring the crucial role of REDD1 in the pathogenesis of meta-inflammation, insulin resistance, and T2D through atypical NF-κB activation.

Taken together, our findings provide evidence that HFD-induced REDD1 stimulates obesity and insulin resistance through cell-type-specific functions, such as adipogenesis and macrophage inflammation, in adipose tissue through the atypical activation of NF-κB (Supplementary Fig. 13). REDD1-dependent NF-κB activation is stimulated through IKK-independent NF-κB activation by sequestering IκBα, which masks the NLS of NF-κB p65 and keeps it in an inactive state in the cytoplasm. Thus, we propose that the REDD1–NF-κB axis is the molecular link between obesity-related inflammation and insulin resistance and should be targeted in therapeutic strategies to normalize body weight and improve metabolic complications. Because REDD1-induced expression of proinflammatory cytokines, including TNF-α and IL-1, can subsequently promote the canonical NF-κB pathway, the functional involvement of IKKβ in HFD-induced adipogenesis, meta-inflammation, and glucose metabolism dysregulation cannot be ruled out[7]. We speculate that REDD1 is extremely important but not indispensable for promoting obesity-induced characteristics. Therefore, studies confirming the crosstalk between REDD1-dependent atypical and IKKβ-dependent canonical mechanisms of NF-κB activation in the pathogenesis of obesity-related metabolic complications are warranted.

## Methods

### Mouse models

All mouse experiments were approved by the Animal Ethics Committee of Kangwon National University (approval numbers KW-190425-5, KW-190723-1, KW-190819-1, and KW-190819-1). All experiments were conducted adhering to the NIH Guide for the Care and Use of Laboratory Animals. Male C57BL/6, *ob/ob*, and *db/db* mice were obtained from Orient Bio Inc. (Sungnam, South Korea). *Redd1*+/− mice were bred and maintained in the animal facility at Kangwon National University and intercrossed to obtain *Redd1*−/− and WT littermates as previously described[43]. *Redd1*flox/flox (*Redd1*fl/fl) and *Redd1*Lys219Ala/Lys220Ala knock-in (*Redd1*KKAA) mice were generated using CRISPR/Cas9 in C57BL/6 zygotes (ToolGen Inc. Seoul, South Korea), and their genotypes were confirmed by PCR analysis and genomic DNA sequencing (Supplementary Figs. 14 and 15). *Adipoq*-Cre and *LysM*-Cre mice were purchased from the Jackson Laboratories (Bar Harbor, ME, USA). *Redd1*fl/fl mice were crossed with *Adipoq*-Cre or *LysM*-Cre mice to generate mice with adipocyte- (*Redd1*ΔAdipoq) or myeloid-specific deletion of *Redd1* (*Redd1*ΔLysM), respectively. Mice were housed in a specific pathogen-free facility (12 h light/dark cycle) that was maintained at 18–24 °C and 30–50% humidity. Six-week-old male mice were fed either NC (10% calories as fat, #D12450B, JA BIO, Inc., Suwon, South Korea) or HFD (60% calories as fat, #D12492, JA BIO, Inc.) for 10 weeks to analyze expression levels of adipogenic genes *Pparg*, *Cebpa*, and *aP2* or for 16 weeks to examine obesity and metabolic complications. Food consumption was monitored daily and body weight was recorded weekly.

### Metabolic parameter measurements

Fasting blood glucose and plasma insulin levels in mice fed NC or HFD for 16 weeks were measured after fasting for 12 and 6 h using a glucometer (Accu-Check Performa, Roche Diagnostics, Mannheim, Germany) and a mouse enzyme-linked immunosorbent assay kit according to the manufacturer's instructions (#M1104, Morinaga, Yokohama, Japan), respectively. HOMA-IR scores were calculated using the following formula[65]: HOMA-IR = serum insulin (mmol/L) × (blood glucose (mmol/L) / 22.5. For GTT, mice were first subjected to fasting for 12 h, and then, intraperitoneally (i.p.) injected with glucose (2 g/kg body weight). Glucose levels were measured from whole blood samples taken from the tail vein at the indicated time using the Accu-Check Performa glucometer. For ITT, mice were i.p. injected with insulin (0.75 unit/kg body weight, Lilly, Indianapolis, IN, USA) after 4 h of fasting, and blood glucose levels were measured as described above.

### Blood and tissue isolation

At the end of the experiments, mice fed NC or HFD for 16 weeks were euthanized using $CO_2$. Blood samples were collected through a cardiac puncture in EDTA-treated tubes on ice unless otherwise specified, and multiple tissues, including the eWAT, iWAT, pancreas, gastrocnemius skeletal muscle, and liver, were harvested, weighed, snap-frozen in liquid nitrogen, and stored at −80 °C until processing. Plasma was collected from anticoagulated whole blood after centrifugation at 2000 × *g* for 20 min at 4 °C and stored at −80 °C until further analysis. All the tissues and plasma were used for biochemical and histological analyses. Expression levels of the adipogenic genes *Pparg*, *Cebpa*, and *aP2* in the eWAT were evaluated in mice fed NC or HFD for 10 weeks.

### Adipocyte differentiation

Adipose SVF cells were isolated from eWAT pads of 7-week-old male C57BL/6 J WT and *Redd1*−/− mice. The eWAT pads were excised from the mice, rinsed with Hanks' balanced salt solution (HBSS, Sigma-Aldrich), finely cut, and incubated in HBSS containing 0.2% collagenase type 2 (Worthington, Lakewood, NJ, USA) at 37 °C for 60 min with agitation (60 cycle/min). The reaction was stopped by adding fetal bovine serum (FBS, Logan, UT, USA) to a final concentration of 10%, The digested tissue was filtered through a 100-μm cell strainer and the filtrate was centrifuged at 500 × *g* at 4 °C for 5 min. The upper lipid phase was removed, and the aqueous phase with the pellet was filtered through a 70 μM cell strainer and centrifuged at 500 × *g* at 4 °C for 7 min. The cell pellet was resuspended in Dulbecco's modified Eagle's medium (DMEM), containing 15% bovine calf serum (#16170-078, Gibco, Grand Island, NY, USA) and plated on a 100-mm collagen-coated dish. The cells were harvested, replated on a 6 well-plate, and grown to 100% confluence. They were maintained for 48 h post-confluence before differentiation. Cells were incubated in DMEM containing dexamethasone (1 μM, Sigma-Aldrich), isobutylmethylxanthine (0.5 μM, Sigma-Aldrich), insulin (10 μg/mL, #16634, Sigma-Aldrich), and 10% FBS for 2 days. This was followed by treatment with insulin (10 μg/mL) for 2 days and incubation in fresh media for 4 days. In addition, mouse adipose SVF cells and 3T3-L1 preadipocytes (5.0 × 10^5 cells/well, American Type Culture Collection) were plated and transfected with sh*Redd1* (#sc-45807-SH, Santa Cruz Biotechnology, Santa Cruz, CA. USA) or pcDNA3.1/His-*Ikba* (encoding His-tagged IκBα) using Lipofectamine 2000 (Thermo Fisher Scientific, Waltham, MA USA). A day after reaching 100% confluence, the transfected cells were cultured in serum-free media for 2 h and infected with empty adenovirus (Ad-control) or adenovirus expressing mouse *Redd1* (Ad-*Redd1*) or its mutants at a multiplicity of infection (MOI) of 500 in Opti-MEM-reduced serum medium using poly-l-lysine (0.5 μg/mL) for 4 h, followed by incubation in fresh DMEM containing 10% fetal calf serum for 2 days. Cell differentiation was induced by culturing the cells in fresh media containing insulin (10 μg/mL) for 2 days and then maintaining

them in fresh media for 4 days. Differentiated adipocytes were fixed with 4% formaldehyde for 1 h, stained with oil red-O for 2 h at 25 °C, and washed three times with distilled water. The stained lipid droplets were visualized using light microscopy, dissolved in 100% isopropanol, and quantified using an ELISA reader at 500 nm.

## Measurement of cytokines, chemokines, and ALT
The concentrations of TNF-α (#MTA00B), MCP-1 (#MJE00B), IL-1β (#MLB00C), IL-6 (#M6000B), leptin (#MOB00), resistin (#MRSN00), and adiponectin (#MRP300) were measured in the plasma and culture medium using Quantikine ELISA kits (R&D Systems, Minneapolis, MN, USA) according to the manufacturer's protocols. Plasma ALT levels were determined as previously described[18].

## RNA isolation and qRT-PCR
Total RNA was isolated from tissues and cultured cells using the TRIzol reagent (Invitrogen, Carlsbad, CA, USA) or RNeasy kit (Qiagen, Hilden, Germany) and reverse-transcribed into cDNA using the M-MLV Reverse Transcriptase (Promega, Madison, WI, USA) according to the manufacturers' protocols. Gene expression levels were quantified using qRT-PCR with the real-time PCR cycler Rotor-Gene Q (Qiagen) using target gene-specific primers (Supplementary Table 1). Relative mRNA expression was calculated by normalizing the expression level of target genes with that of the housekeeping gene glyceraldehyde-3-phosphate dehydrogenase (Gapdh).

## Western blotting
Samples of WAT, liver, and skeletal muscle were collected from mice 15 min after an i.p. injection of saline or insulin (5 mU/g body weight) and quickly frozen in liquid nitrogen. Tissue lysates were prepared with RIPA buffer (Elpis Biotech, Daejeon, Korea), and membrane fractions were isolated using the Subcellular Protein Fractionation Kit (#87790, Thermo Fisher Scientific). Samples were separated by 10–15% SDS-PAGE and transferred onto polyvinylidene difluoride membranes (Millipore Sigma). The membranes were blocked with 3% bovine serum albumin in Tris-buffered saline (TBS)−0.1% Tween 20 buffer (0.1% TBST buffer, pH 7.4) for 1 h and immunoblotted with target protein-specific antibodies, followed by visualization of the target protein bands using an HRP-conjugated goat anti-rabbit IgG secondary antibody (#31460, Thermo Fisher Scientific). Antibodies against phospho-Akt (Thr$^{308}$, #9275), Akt (#9272), phospho-IRS-1 (Tyr$^{895}$, #3070), IRS-1 (#2382), GLUT4 (#2299), phospho-FOXO1 (Ser$^{253}$ in mouse/Ser$^{256}$ in human, #84192), FOXO1(#2880) phospho-S6 (Ser$^{240/244}$, #2211), S6 (#2217), phospho-S6K (Thr$^{389}$, #9234), S6K (#34475), phospho-S6 (Ser$^{240/244}$, #2211), S6 (#2217), phospho-IKKαβ (Ser$^{176/180}$, #2697), IKKβ (#2684), PPARγ (#2435), CEBPα (#2295), PARP (#9542), IκBα (#9242), β-catenin (#9562), and His-Tag (#2365) were purchased from Cell signaling technology (Danvers, MA, USA). Antibodies for REDD1 (#10638-1-AP) and NF-κB p65 (#sc-372) were obtained from Proteintech (Rosemont, IL, USA) and Santa Cruz Biotechnology, respectively. Relative protein levels were quantified using ImageJ (NIH, Bethesda, MD, USA).

## Immunohisto-cytochemical examination
The eWAT and pancreas were fixed in 4% formaldehyde at 4 °C for 24 h. The tissues were embedded in paraffin, sectioned to a thickness of 5 μm, deparaffinized, and rehydrated. Adipose tissue sections were incubated with blocking solution (Agilent Dako North America, Carpinteria, CA, USA) for 1 h, followed by overnight incubation with rabbit monoclonal perilipin-1 antibody (1:300, #9349 S, Cell Signaling Technology) and mouse monoclonal F4/80 antibody (1:300, #sc-377009, Santa Cruz Biotechnology) at 4 °C. The sections were washed three times with TBS (pH 7.6) and incubated with Alexa Fluor® 488 goat anti-rabbit (IgG) (1:500, #A-11034, Thermo Fisher Scientific) and Alexa Fluor® 647 goat anti-mouse (1:500, #A-32728, Thermo Fisher

Scientific) secondary antibodies at 4 °C for 2 h. Sections from the pancreas were incubated overnight with rabbit monoclonal insulin antibody (1:250, #3014 S, Cell signaling) at 4 °C, washed three times with TBS buffer, and incubated with Alexa Fluor® 488 goat anti-rabbit secondary antibody (1;1000, #A11034, Thermo Fisher Scientific) for 2 h at 4 °C. Then, the slides were rinsed, incubated with DAPI (1 μg/mL, Sigma-Aldrich, St. Louis, MO, USA) for 30 min at room temperature in the dark, and washed three times. The sections were mounted using the Mounting Medium (Agilent Dako North America Inc., Carpinteria, CA, USA), and images were acquired and analyzed using the Zeiss LSM 880 laser scanning confocal microscope with AiryScan (Zeiss, Oberkochen, Germany). CLS number was determined in the eWAT by counting the number of adipocytes surrounded by F4/80-positive signals per unit area. In addition, liver tissues were embedded in paraffin, sectioned to a thickness of 7 μm, stained with H&E, and imaged on a microscope scanner (Grundium Ocus®40, Tampere, Finland). A score of steatosis was performed using NIH ImageJ software. The nuclear translocation of NF-κB p65 was visualized in human embryonic kidney 293 (HEK293) cells (Korean Cell Line Bank, Seoul, Korea) transfected with the pFlag-CMV-1 vector (#E7273, Sigma-Aldrich) alone or containing Redd1 or its mutants, such as Redd1$^{KKAA}$ or Redd1$^{KKRAAA}$, using Lipofectamine 2000 (Thermo Fisher Scientific).

## Luciferase reporter, Chip, and NF-κB activity assays
3T3-L1 cells were transfected with 80 nM of siRNA against Ikka (encoding IKKα, #sc-29366, Santa Cruz Biotechnology), Ikkb (encoding IKKβ, #sc-35645), or NF-κB p65 (#sc-29411) as well as pcDNA3.1/His vector or pcDNA3.1/His-Ikba using Lipofectamine 2000 (Thermo Fisher Scientific), as described[18]. Mouse peritoneal macrophages were isolated from WT mice, as described[66]. Macrophages or 3T3-L1 cells (1 × 10$^6$ cells/well) were co-transfected with 0.5 μg of p5×NF-κB−Luc (Stratagene, La Jolla, CA, USA) or PGL3 vector (Promega) containing approximately 2.8, 2.1, or 1.4 kbp upstream of the Cepba translation start site and 0.5 μg of the Renilla luciferase reporter vector pGL4.74 vector (Promega) as a control reporter using Lipofectamine 2000 (Thermo Fisher Scientific) and stabilized in fresh medium containing 10% FBS for 24 h. The cells were infected with Ad-control or Ad-Redd1 at an MOI of 100 for 6 h, followed by incubation in a fresh medium for 24 h. The cell lysates were prepared using ice-cold lysis buffer (#E1941, Promega). Luciferase activity was measured using a Dual Luciferase Assay kit (#E1960, Promega). Firefly luciferase activity was normalized to Renilla activity. Chip assay was performed in 3T3-L1 cells infected with Ad-control or Ad-Redd1 using primers detailed in Supplementary Table 2 as previously described[67]. NF-κB activity was assayed in tissue lysates using an NF-κB/DNA-binding ELISA kit (#OKAG00423, Aviva Systems Biology, San Diego, CA, USA) according to the manufacturer's instructions.

## Co-immunoprecipitation
HEK293 cells were transfected with pcDNA3.1/His vector alone or containing Ikba and pFLAG-CMV-1 vector alone or containing Redd1, Redd1$^{KKAA}$, or Redd1$^{KKRAAA}$ using Lipofectamine 2000. Cells were rinsed two times with ice-cold phosphate-buffered saline and lysed in modified RIPA buffer [100 mM Tris-HCl (pH 7.6), 5 mM EDTA, 50 mM NaCl, 50 mM β-glycerophosphate, 50 mM NaF, 0.1 mM Na$_3$VO$_4$, 1% Brij-35, 0.5% sodium deoxycholate, and 1 mM phenylmethylsulphonyl fluoride] for 30 min on ice. After centrifugation for 15 min at 12,000 × g at 4 °C, the supernatant was collected. Total protein (1 mg) was precleared with protein G plus/protein A-agarose (Millipore, 50% slurry in modified RIPA buffer) for 1 h at 4 °C. The indicated antibodies (1–3 μg) were added to 500 μL of precleared protein extracts. The precleared protein extracts (500 μL) were mixed with 2 μg of anti-FLAG M2 antibody (#F3165, Sigma-Aldrich) or anti-His-Tag antibody (#2365, Cell Signaling) and rotated for 24 h at 4 °C, as described[18]. Then, 30 μL of

protein A/G plus agarose beads (50% slurry) were added to the mixture, which was rotated at 4 °C for 2 h. Immunoprecipitates were collected by centrifugation and washed with modified RIPA buffer three times. Samples were boiled for 10 min with a loading buffer. Immune complexes were separated using SDS-PAGE (10–15%) and subjected to immunoblot analysis.

### Isolation of adipocytes, SVF cells, and adipose tissue macrophages

eWAT of WT and $Redd1^{-/-}$ mice fed HFD for 16 weeks were minced, chopped, and digested with collagenase II (1 mg/ml, Sigma-Aldrich, St. Louis, MO, USA; #C2674) in Krebs-Ringer HEPES Buffer for 15 min at 37 °C with gentle shaking. After passing cells through a 200 µm cell strainer and centrifugation at 1000× g for 10 min, the floating cells were collected as the mature adipocytes, and the pelleted cells were obtained as the SVF cells. The adipocytes were washed twice with DMEM to remove cell debris by centrifugation as above. The SVF cells were resuspended in erythrocyte lysis buffer and incubated at room temperature for 5 min. The erythrocyte-depleted SVF cells were centrifuged at $500 \times g$ for 5 min and the pellet was resuspended and used for positive selection of adipose tissue macrophages. Briefly, the cells were labeled with the CD11b Microbead (10 µl/$10^7$ cells, #130-049-601, Miltenyi Biotec Inc., Bergisch Gladbach, Germany) and passed through magnetic-activated cell sorting (MACS) separation column (Miltenyi Biotec). After washing twice, CD11b$^+$ cells were removed from the column by washing twice with 2 ml MACS buffer away from the magnetic field. The purity of the isolated cells was determined using FITC-conjugated anti-CD11b (#11-0118-42, eBioscience, San Diego, CA) and PE-conjugated anti-F4/80 antibodies (#15-4801-82, eBioscience) and was greater than 95%.

### Protein-protein docking simulations

HADDOCK (https://wenmr.science.uu.nl) and HDOCK (https://hdock.phys.hust.edu.cn/) were used to predict a binding conformation between the two proteins. Structures of the NF-κB–IκB complex (PDB ID: 1NFI) and REDD1 (PDB ID: 3LQ9) were obtained from the RCSB protein databank (https://www.rcsb.org) and used for protein-protein docking simulations. These protein structures were protonated at a physiological pH using the clean protein protocol of Discovery Studio 2018. In molecular docking calculations, IκB was assigned as the first molecule with the active residues, Asp$^{74}$, Asp$^{75}$, and Glu$^{85}$, which interact with Arg$^{304}$, Lys$^{301}$, and Arg$^{302}$ of NLS of NF-κB p65, respectively[30]. REDD1 was submitted as the second molecule with active lysine residues at positions 218, 219, and 220, corresponding to the NLS of p65.

### Statistics and reproducibility

Statistical analysis was performed using GraphPad Prism (version 6.0; GraphPad Software Inc.). No statistical methods were used to predetermine the sample size. The experiments were randomized, investigators were blinded to allocation during experiments and outcome analysis, and no samples or animals were excluded from the analysis. The number of replicates are listed in the figures and the representative results were obtained from at least three independent experiments. All values are presented as the mean ± standard error of the mean (s.e.m.). Statistical significance was determined by an unpaired two-tailed $t$-test between two groups or one-way or two-way ANOVA followed by the Holm−Sidak multiple comparisons test, depending on the experimental groups analyzed. Statistical significance was set at $P < 0.05$.

### Reporting summary

Further information on research design is available in the Nature Research Reporting Summary linked to this article.

## Data availability

The data supporting the findings of this study are available within the article and its Supplementary Information files. Uncropped blots and all the data used to generate the graphs of the main and supplementary figures can be found in the Source Data file. Source data are provided with this paper. Structures of the NF-κB–IκB complex (PDB ID: 1NFI) and REDD1 (PDB ID: 3LQ9) were obtained from the RCSB protein databank (https://www.rcsb.org). Source data are provided with this paper.

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

## Acknowledgements

This work was supported by the National Research Foundation of Korea (NRF) Grant funded by the Korea Government (MSIP) (NRF-2017R1A2B3004565 and NRF-2022R1A2C3005747 to Y.M.K.; NRF-2020R1A5A8019180 to J.H.L.).

## Author contributions

D.K.L., T.K., and Y.M.K. conceptualized the study and designed all experiments. D.K.L., T.K., J.B., and S.K. conducted the experiments and collected and analyzed the data. J.K. and S.C. assisted with the experimental analyses. G.L., C.P., K.W.L., and Y.J.K. carried out protein-protein docking modeling. M.P. performed the statistical analysis. D.K.L. and Y.M.K. wrote the manuscript in discussion with J.H.L. and Y.G.K., who provided intellectual inputs. Y.M.K. conceived the project and supervised the research.

## Competing interests

The authors declare no competing interests.

## Additional information

**Correspondence and requests** for materials should be addressed to Young-Myeong Kim.

