## [Peer Review File · Nature Communications]

Title: REDD1 promotes obesity-induced metabolic dysfunction via atypical NF- κ B activationREVIEWER COMMENTS

Reviewer #1 (Remarks to the Author):

In the present manuscript, the authors have investigated the implication of REDD1 in the development of obesity and metabolic complications. They used several mouse models with whole-body deletion of REDD1 and specific deletion of REDD1 in adipocytes and myeloid cells. They show that deletion of REDD1 in whole body and in adipocytes protects mice from developing obesity, adipose tissue inflammation and insulin resistance. Deletion of REDD1 in the myeloid compartment does not affect the development of obesity but leads to a protection against insulin resistance induced by obesity.

To decipher the molecular mechanisms involved in the action of REDD1, the authors show that REDD1 stimulates adipocyte differentiation through the activation of NFκB. Mutations of IκBa binding sites prevents NFκB activation in vitro and obesity in vivo.

This study is complete with many mouse models and cellular models. The results presented are interesting and give new information concerning the implication of REDD1 in obesity-induced metabolic complications.

Several controls, but important points, must be addressed by the authors to complete their study.

1. The effects detected with REDD1^{-/-} mice (development of obesity, glucose metabolism, GTT, ITT, HOMA-IR, cytokines expression) are strong and convincing. How the authors explain their observations compared to a publication in which deletion of REDD1 did not affect the development of obesity (PMID 32043636).
2. How REDD1^{-/-} and REDD1^{fl/fl} mice were generated? Which exon have been deleted? Where loxP sites have been inserted? The authors should give additional information to compare the mice used in this study to the mice used in previous studies.
3. The authors use WT mice as control to compare with the results obtained with REDD1^{-/-} mice? What are these WT control mice ? commercially available WT mice ? Littermates obtained from REDD1^{+/-} breeding ? This point must be clarified.
4. The authors show that REDD1 expression is increased in liver, muscle and eWAT of obese mice (HFD, ob/ob and db/db). What is the expression pattern of REDD1 in adipocytes and stroma vascular fraction during obesity? is REDD1 induced only in adipocytes, or also in adipose tissue macrophages?
5. The authors must show the protein expression of REDD1 in WT and REDD1^{-/-} mice in NC and HFD in eWAT, liver and skeletal muscle (Figure 1 and 2).
6. In figures 1 to 3, the authors investigate the insulin signaling pathway in muscle and liver. Why not in eWAT? The authors must investigate insulin signaling pathway in eWAT.
7. Since REDD1 inhibits mTORC1 activation, the authors must investigate mTORC1 signaling pathway (S6K, S6 phosphorylation) in liver, muscle and eWAT.
8. What is the level of hepatic steatosis in all the mouse model used in the present study, since it has been shown that REDD1 deletion decreases hepatic steatosis in HFD (PMID 32043636)?
9. In REDD1^{Δadipoq} mice, the authors must show the expression pattern of REDD1 in whole adipose tissue, adipocytes and stroma vascular fractions, as well as liver and skeletal muscle of REDD1^{fl/fl} and REDD1^{Δadipoq} mice.

10. Does deletion of REDD1 in adipose tissue protects mice from the development of metabolic complications such as hepatic steatosis ?
11. What is the insulin signaling pathway in eWAT of REDD1^{Δadipoq}? What is the mTORC1 signaling in liver, skeletal muscle and eWAT (S6K, S6)? In Figure 3, the authors must show the results of insulin signaling in REDD1^{Δadipoq} in NC condition.
12. Is REDD1 expression increased during adipocyte differentiation ? what about I κ B expression and NF κ B activation during adipocyte differentiation ? Is NF κ B activated in the adipose tissue of obese mice (HFD, ob/ob)? What is the status of NF κ B activity in the adipose tissue of REDD1^{-/-}, REDD1^{Δadipoq} and REDD1^{ΔLysM} ?
13. The authors should show the expression of REDD1, I κ B, p65, in all experimental conditions of figure 5.
14. What is the expression of I κ B after REDD1 expression ? Is I κ B been sequestered and degraded ?
15. In Figure 6 d, c, expression of REDD1 and REDD1^{KKAA} must be shown to be sure that the level of expression is the same
16. In Figure 7, what is the expression level of REDD1 ^{KKAA} in HFD mice ? as well as NF κ B activity, I κ B expression and mTORC1 activity ?

Reviewer #2 (Remarks to the Author):

Completely fails to attract readers due to lack of novelty.
see for the reference.

Sci Rep

. 2017 Aug 1;7(1):7023. doi: 10.1038/s41598-017-07182-z.

Implication of REDD1 in the activation of inflammatory pathways.

Authors fail to show raw data instead of fair data in source of files.

The manuscript is full of severe major inaccuracies and is not suitable for nature communications.

Reviewer #3 (Remarks to the Author):

In this manuscript, the authors investigated the role of REDD1 in obesity and metabolic disorders. The authors utilized several different mouse models including whole body knock out, and conditional knockout mice to study how Redd1 in different cell types regulates obesity and insulin resistance. For example, they found that adipose specific Redd1 knock out alleviated obesity, meta-inflammation, macrophage infiltration, and insulin resistance. Myeloid specific Redd1 knockout only alleviated macrophage infiltration, meta-inflammation, and insulin resistance but not obesity/adiposeness. Additionally, the authors demonstrate possible interactions with Lys219/220 on Redd1 and Glu85 and Asp73 on I κ B α . The animal experiments seem to be well executed and their findings may increase our

understanding of the function of REDD1 in obesity and metabolic disorders. However, the study is relatively descriptive and lack mechanistic insight. The interaction between Redd and I κ B α cannot fully explain the phenotype of the mice. Further, adiponectin-cre only deleted REDD1 in mature adipocyte but not in adipose progenitors, deficiency of adipocyte REDD1 should not affect adipocyte differentiation. Thus, the conclusion and the proposed model in Extended Data Fig 12 is not correct. I have the following specific comments and suggestions:

1. The authors studied adipocyte-specific REDD knockout mice and found that deficiency of adipocyte REDD1 inhibited diet-induced obesity and insulin resistance. However, the underlying mechanisms were not provided. Since adiponectin-cre only deleted REDD1 in mature adipocyte but not adipose progenitors, deficiency of adipocyte REDD1 should not affect adipocyte differentiation. So the conclusion and the proposed model in Extended Data Fig 12 is not correct. More detailed mechanistic studies should be performed to explain this phenotype.

2. The authors found that REDD1 can directly interact with I κ B α but this may only explain the increased inflammation phenotype but not increased adipocyte inflammation. NF- κ B signaling plays complex role in regulating adipocyte differentiation and NF- κ B itself does not directly regulate PPAR γ and CEBP α transcription. The observed adipogenesis effects may not due to NF- κ B's transcriptional activity but other proteins associated with NF- κ B signaling (e.g. IKK, Wnt). More experiments on this topic should be performed.

3. The authors mentioned that mice were on HFD for 10 or 16 weeks. Please specify the groups that were placed for duration of HFD. What's the reason for different feeding duration for the studies?

4.

5. The authors provided data for insulin signaling in liver and skeletal muscle. Since obesity is a major focus of this paper, the authors should also include some insulin signaling data in adipose tissue.

6. The authors state (line 354) that Redd1-induced MCP-1 production in adipocytes is an important player in macrophage activation. Additionally, results show Redd1 signaling in macrophages can similarly induce pro-inflammatory activation. Is Redd1 signaling downregulated or affected in macrophages of Redd1 Δ Adipoq? Can MCP-1 stimulate Redd1 signaling?

7. Previous data suggest IL-6 as a major player in regulating insulin signaling (Cai, 2005, Nature Medicine 11, 183-190). The authors should provide more discussion and data related to IL-6, for example:

- a. Provide data for IL-6 levels in plasma and tissue from the Redd1 Δ LysM model
- b. Provide data for IL-6 target genes in different tissues (e.g. SOCS and Stat)
- c. More discussion on IL-6 role in insulin signaling and possibly move IL-6 data to main figures

8. The authors should also rephrase some claims that are made in the results section to avoid misinterpretation. Since the authors chose to show both NC and HFD groups, they need to clarify which group they are referring to when making a claim. For example:

- a. Line 136: Be sure to state that the reduction in hepatic gluconeogenesis occurs in the HFD group.

9. In line with the previous comment, authors should consider placing NC data in supplemental data and moving some data (such as IL-6 and others) to the main text.

10. The authors should also provide data to verify their KO models (e.g. by western blotting, QPCR, or citations).

Reviewer #4 (Remarks to the Author):

I have read the manuscript NCOMMS-21-39925 entitled "The REDD1–NF- κ B axis is crucial for adipogenesis, meta-inflammation, and metabolic dysfunction", submitted to Nature Comments for publication.

As requested, I will address my main contribution to molecular docking and molecular dynamics aspects. The computational data presented in the manuscript is highly speculative, at least in how it was introduced here. Therefore, other assays are necessary to make outcomes more reliable.

Molecular docking is a powerful computational technique for finding binding poses and estimating binding free energy between two molecular entities. However, it is highly recommendable to use more than one program to accomplish this task. The authors limited themselves to using only the HADDOCK server, a well-known protein-protein server. In this case, the solution is highly dependent on the residues defined as active.

On the other hand, the best pose may not be unique, and other possibilities must be considered. The list of protein-protein docking servers is long. (Examples are: The ClusPro web server for protein-protein docking, ZDOCK, Hex Protein Docking, to name a few). These other servers could retrieve equally valid poses that might be potential solutions to the complex formation.

Complementary analyses such as calculating the electrostatic potential map for both proteins, Redd1 and IKBa, to investigate complementarity would reinforce the quality of the results.

Furthermore, there is no quantitative estimation of the binding intensity. No measure of the interaction energy or even a rough estimate of the binding ΔG using a server as PISA (or equivalent) is given. This would be of particular value when investigating docking between mutated Redd1 protein and the receptor. This quantitative measure must be estimated for the wild and mutated complexes and have to be correlated with experimental data is possible.

Finally, molecular docking is necessary but is not sufficient if taken isolatedly. Although the authors claim they used molecular dynamics simulations in their work, no description of any simulation result was found in the manuscript. Moreover, no dynamic information is given. For example, there is no description of salt bridges occupancies, or hydrogen bonds mean half-lives, increasing the results. Furthermore, it is mandatory to monitor complex stability from docking pose. The system's dynamical evolution provides a clue on the pose's veracity. Badly docked molecules do not remain bound for longer times. Wrong poses will undo in times less than dozens of nanoseconds. So, RMSD of the complex, taking the initial conformation as a reference, will show whether the pose was kept over the

simulation time. Simulations must be carried out in triplicates for longer times than the one simulated here. It is advisable to set three different simulations of about 100 ns minimum for each complex, given that the complex involves heavy macromolecular moieties.

Response to reviewers' comments

Reviewer #1 (Remarks to the Author):

In the present manuscript, the authors have investigated the implication of REDD1 in the development of obesity and metabolic complications. They used several mouse models with whole-body deletion of REDD1 and specific deletion of REDD1 in adipocytes and myeloid cells. They show that deletion of REDD1 in whole body and in adipocytes protects mice from developing obesity, adipose tissue inflammation and insulin resistance. Deletion of REDD1 in the myeloid compartment does not affect the development of obesity but leads to a protection against insulin resistance induced by obesity.

To decipher the molecular mechanisms involved in the action of REDD1, the authors show that REDD1 stimulates adipocyte differentiation through the activation of NFkB. Mutations of Ikb α binding sites prevents NFkB activation in vitro and obesity in vivo. This study is complete with many mouse models and cellular models. The results presented are interesting and give new information concerning the implication of REDD1 in obesity-induced metabolic complications.

Several controls, but important points, must be addressed by the authors to complete their study.

1. The effects detected with *Redd1*^{-/-} mice (development of obesity, glucose metabolism, GTT, ITT, HOMA-IR, cytokines expression) are strong and convincing. How the authors explain their observations compared to a publication in which deletion of REDD1 did not affect the development of obesity (PMID 32043636).

Response: As per your comment, we have now compared our results with previously reported observations including that by Dumas et al. in the Discussion section as follows. "There are a few studies describing the role of REDD1 in metabolic dysregulation and hepatic steatosis in HFD-induced obese mouse models^{16,20}. Williamson et al. reported that *Redd1*^{-/-} mice showed reduced weight gain and lower blood glucose levels than WT mice under HFD conditions¹⁶, which is consistent with our findings. On the other hand, Dumas et al. demonstrated that in HFD-fed mice,

Redd1 deficiency prevented hepatic steatosis by decreasing the expression of lipogenic enzymes including SREBP-1c, FASN, and SCD-1, without affecting weight gain and glucose intolerance¹⁶. These results are in part different from our findings that global or adipocyte-specific *Redd1* deletion protected mice from HFD-induced obesity and hepatic steatosis. In general, chronic lipid overload, such as HFD feeding, elicits lipid redistribution between adipose tissue and liver, resulting in a similar metabolic phenotype in both organs through lipid homeostasis; however, the contrasting effects (protective or non-protective) of *Redd1* deficiency on metabolic dysregulation i.e., obesity in an HFD-fed mouse model could arise from complex interactions among genetic background, nutrient composition, and environmental stress^{43–46}.”

2. How *Redd1*^{-/-} and *Redd1*^{fl/fl} mice were generated? Which exon have been deleted? Where loxP sites have been inserted? The authors should give additional information to compare the mice used in this study to the mice used in previous studies.

Response: We have now provided additional information on genetically modified mice used in this study in the Methods section as follows. “Male C57BL/6, *ob/ob*, and *db/db* mice were obtained from Orient Bio Inc. (Sungnam, South Korea). *Redd1*^{+/-} mice were bred and maintained in the animal facility at Kangwon National University and intercrossed to obtain *Redd1*^{-/-} and WT littermates as previously described⁴². *Redd1*^{flox/flox} (*Redd1*^{fl/fl}) and *Redd1*^{Lys219Ala/Lys220Ala} knock-in (*Redd1*^{KKAA}) mice were generated using CRISPR/Cas9 in C57BL/6 zygotes (ToolGen Inc. Seoul, South Korea), and their genotypes were confirmed by PCR analysis and genomic DNA sequencing (Supplementary Figs. 14 and 15). *Adipoq*-Cre and *LysM*-Cre mice were purchased from the Jackson Laboratories (Bar Harbor, ME, USA). *Redd1*^{fl/fl} mice were crossed with *Adipoq*-Cre or *LysM*-Cre mice to generate mice with adipocyte- (*Redd1*^{Δ*Adipoq*}) or myeloid-specific deletion of *Redd1* (*Redd1*^{Δ*LysM*}), respectively.”

3. The authors use WT mice as control to compare with the results obtained with *Redd1*^{-/-} mice? What are these WT control mice? Commercially available WT mice?

Littermates obtained from *Redd1*^{+/-} breeding? This point must be clarified.

Response: As recommended by the reviewer, we have now provided more information on mice used in this study in the Methods section as follows. “*Redd1*^{+/-} mice were bred and maintained in the animal facility at Kangwon National University and intercrossed to obtain *Redd1*^{-/-} and WT littermates as previously described⁴².”

4. The authors show that REDD1 expression is increased in liver, muscle and eWAT of obese mice (HFD, *ob/ob* and *db/db*). What is the expression pattern of REDD1 in adipocytes and stroma vascular fraction during obesity? is REDD1 induced only in adipocytes, or also in adipose tissue macrophages?

Response: As per the reviewer’s suggestions, we isolated mature adipocytes, stromal vascular fraction cells, and macrophages from eWAT of C57BL/6 mice fed NC or HFD. Expression levels of *Redd1* were determined in these cells by qRT-PCR analysis. We have now added the following sentence and the data to the Results section and Supplementary Fig. 1b respectively. “In addition, REDD1 expression also increased in adipocytes, stromal vascular fraction (SVF) cells, and adipose tissue macrophages isolated from eWAT of HFD-fed C57BL/6 mice (Supplementary Fig. 1b).”

5. The authors must show the protein expression of REDD1 in WT and *Redd1*^{-/-} mice in NC and HFD in eWAT, liver and skeletal muscle (Figures 1 and 2).

Response: We determined the expression levels of REDD1 in the eWAT, liver, and skeletal muscle of WT and *Redd1*^{-/-} mice that were fed NC or HFD. We have now added the data to Supplementary Fig. 1c and the following sentence to the Results section. “Therefore, we investigated the role of REDD1 as a regulator of adipogenesis and obesity using *Redd1*^{-/-} mice, which were confirmed by depletion of REDD1 expression in the eWAT, skeletal muscle, and liver after being fed HFD (Supplementary Fig. 1c).”

6. In figures 1 to 3, the authors investigate the insulin signaling pathway in muscle and

liver. Why not in eWAT? The authors must investigate insulin signaling pathway in eWAT.

Response: We additionally examined the insulin signaling in adipose tissue of *Redd1^{-/-}*, *Redd1^{ΔAdipoq}*, *Redd1^{ΔAdipoq}*, *Redd1^{KKAA}* and control (WT and *Redd1^{fl/fl}*) mice fed NC or HFD and found that insulin signaling was rescued in adipose tissues, as shown in other tissues such as liver and skeletal muscle of HFD-fed *Redd1^{-/-}*, *Redd1^{ΔAdipoq}*, and *Redd1^{ΔLysM}* mice, but not in NC-fed mouse groups. We have now added these results to Figures 2f, 3i, 4i, and 7j.

7. Since REDD1 inhibits mTORC1 activation, the authors must investigate mTORC1 signaling pathway (S6K, S6 phosphorylation) in liver, muscle and eWAT.

Response: We examined levels of S6K and S6 phosphorylation in the liver, skeletal muscle, and eWAT of *Redd1^{-/-}*, *Redd1^{ΔAdipoq}*, and *Redd1^{KKAA}*, and control mice fed NC or HFD as well as REDD1-overexpressing 3T3L-1 preadipocytes. The results were shown in Supplementary Fig. 12a–d and the following sentences were added to the Discussion section, “Notably, HFD increased phosphorylation of ribosomal protein S6 kinase (S6K) and ribosomal protein S6, the downstream targets of mTORC1, in the liver, skeletal muscle, and eWAT of *Redd1^{-/-}*, *Redd1^{ΔAdipoq}*, *Redd1^{KKAA}*, and control mice; however, the phosphorylation levels were not different between *Redd1*-targeted mice and control littermates under the same feeding conditions (Supplementary Fig. 12a–c). These observations were consistent with previous results showing a similar increase of mTORC1 activity in skeletal muscle of both HFD-fed *Redd1^{-/-}* and WT mice²⁰, but different from other results^{16,40,41}. For instance, mTORC1 activity was unexpectedly decreased in the liver of HFD-fed *Redd1^{-/-}* mice¹⁶ and unchanged in *Redd1*-disrupted hepatocyte and *Redd1*-overexpressing C3H10T1/2 adipocytes^{40,41}. Interestingly, we found that REDD1 overexpression inhibited the mTORC1 signaling pathway in cultured endothelial cells⁴² and 3T3-L1 preadipocytes (Supplementary Fig. 12d). Although the mechanisms involved remain unknown, our findings suggest that REDD1 promotes rather than prevents obesity-induced metabolic disorders in an mTORC1-independent manner.”

8. What is the level of hepatic steatosis in all the mouse model used in the present study, since it has been shown that REDD1 deletion decreases hepatic steatosis in HFD (PMID 32043636)?

Response: As suggested by the reviewer, we investigated the role of REDD1 in hepatic steatosis using *Redd1*^{-/-}, *Redd1*^{ΔAdipoq}, *Redd1*^{ΔLysM}, *Redd1*^{KKAA}, and control mice, as well as serum levels of alanine aminotransaminase and expression levels of hepatic lipogenic genes. The results were added to Fig. 8 and Supplementary Fig. 11 and the following paragraph was added to the Results section. **“Global or adipocyte-specific loss of *Redd1* prevents HFD-induced hepatic steatosis.** Since obesity is associated with fatty liver, we examined the role of REDD1 in HFD-induced hepatic steatosis. HFD feeding resulted in effectively developed hepatic steatosis in WT or *Redd1*^{fl/fl} mice, as evidenced by H&E staining of the liver tissues, and the characteristic features of steatosis were significantly prevented in *Redd1*^{-/-}, *Redd1*^{ΔAdipoq}, and *Redd1*^{KKAA} mice, consistent with a recent study¹⁶, but not in *Redd1*^{ΔLysM} mice; however, no hepatic steatosis was observed in NC-fed mice (Fig. 8a and Supplementary Fig. 11a). The histological features of steatosis are highly correlated with circulating levels of alanine aminotransaminase (ALT), an index of hepatic injury (Supplementary Fig. 11b). In addition, the expression of hepatic lipogenic genes, including acetyl-CoA carboxylase (*Acc*), fatty acid synthase (*Fasn*), and stearoyl-CoA desaturase-1 (*Scd-1*), were decreased in *Redd1*^{-/-}, *Redd1*^{ΔAdipoq}, and *Redd1*^{KKAA} mice, but not in *Redd1*^{ΔLysM} mice on HFD (Fig. 8b). This indicates that REDD1 regulates lipogenesis and lipid homeostasis by the interplay between metabolically active organs such as liver and adipose tissue, but not immune cells. Thus, our findings suggest that ablation of *Redd1* in the whole body or adipocytes, but not in myeloid cells, prevents HFD-induced hepatic steatosis by suppressing atypical NF-κB-dependent lipogenesis rather than meta-inflammation.”

9. In *Redd1*^{ΔAdipoq} mice, the authors must show the expression pattern of REDD1 in whole adipose tissue, adipocytes and stromal vascular fractions, as well as liver and skeletal muscle of *Redd1*^{fl/fl} and *Redd1*^{ΔAdipoq} mice.

Response: We examined the REDD1 expression pattern in various tissues of *Redd1^{fl/fl}* and *Redd1^{ΔAdipoq}* mice by western blotting and added new data and the following sentence to Supplementary Fig. 1d and the Results section, respectively. “we examined the role of adipocyte REDD1 in HFD-induced adipogenesis and inflammation using *Redd1^{ΔAdipoq}* mice that show the specific deletion of REDD1 in mature adipocytes, but not in the liver and skeletal muscle (Supplementary Fig. 1d).”

10. Does deletion of *Redd1* in adipose tissue protects mice from the development of metabolic complications such as hepatic steatosis?

Response: We examined the role of REDD1 in the development of hepatic steatosis using *Redd1^{-/-}*, *Redd1^{ΔAdipoq}*, *Redd1^{ΔLysMq}*, and *Redd1^{KKAA}* mice and added the results to Figure 8 and Supplementary Fig. 11. In addition, a new paragraph was added to the Result section as indicated in the response to comment #8.

11. What is the insulin signaling pathway in eWAT of *Redd1^{ΔAdipoq}*? What is the mTORC1 signaling in liver, skeletal muscle and eWAT (S6K, S6)? In Figure 3, the authors must show the results of insulin signaling in REDD1^{ΔAdipoq} in NC condition.

Response: We examined the insulin signaling in adipose tissue of *Redd1^{fl/fl}* and *Redd1^{ΔAdipoq}* mice fed NC or HFD and found that insulin signaling was improved in adipose tissues, as shown in the liver and skeletal muscle, of HFD-fed *Redd1^{ΔAdipoq}* mice compared to that in the WT littermate controls. The results were added to Figure 2i. In addition, phosphorylation of S6K and S6 was examined using western blot analysis. The results and the following sentence were added to Supplementary Fig. 12b and the results were described in the Discussion section as indicated in the response to comment #7.

12. Is REDD1 expression increased during adipocyte differentiation? what about IκB expression and NF-κB activation during adipocyte differentiation? Is NF-κB activated in the adipose tissue of obese mice (HFD, *ob/ob*)? What is the status of NF-κB activity

in the adipose tissue of *Redd1*^{-/-}, *Redd1*^{Δadipoq} and *Redd1*^{ΔLysM}?

Response: We found that NF-κB activation/activity was lower in the eWAT of *Redd1*^{-/-}, *Redd1*^{Δadipoq} and *Redd1*^{ΔLysM}, and *Redd1*^{KKAA} mice than in the control mice under HFD conditions, but not in the NC-fed mouse groups. These results were added to Figs. 1g, 3d, 4d, and 7e as well as Supplementary Figs. 4h, 6j, and 10i. We also found that REDD1 expression, nuclear NF-κB p65 translocation, and NF-κB activation/activity were increased without affecting IκBα levels in the exposed differentiation media (MDI) and REDD1-overexpressing 3T3-L1 preadipocytes. The results were added to Supplementary Fig. 8a, b, e, f. In addition, we added the following sentences to the Results section. “We next examined whether NF-κB activation can be stimulated during adipocyte differentiation. When cultured in a differentiation medium containing MDI, WT adipose SVF cells and 3T3-L1 cells showed increased REDD1 expression, NF-κB p65 nuclear translocation, and NF-κB-reporter activity without affecting IκBα levels, as shown by REDD1 overexpression (Supplementary Fig. 8a, e, f). This suggests that the REDD1–NF-κB axis plays an important role in adipocyte differentiation, consistent with previous studies showing that NF-κB plays an important role in adipogenic differentiation^{24,25}.”

13. The authors should show the expression of REDD1, IκBα, p65, in all experimental conditions of Figure 5.

Response: As per the reviewer’s suggestion, we determined expression levels of all the target genes in 3T3-L1 cells and macrophages transfected with shRNAs or siRNAs and those infected with Ad-*Redd1* and have added the data to Supplementary Fig. 8a-d, j.

14. What is the expression of IκB after REDD1 expression? Is IκB been sequestered and degraded?

Response: We have previously demonstrated that overexpressed *Redd1* interacts with IκB in RAW264.7 cells without altering IκB levels and stimulates NF-κB activation

and proinflammatory cytokine expression (ref.18). This suggests that *Redd1* elicits interactive sequestration of I κ B and subsequent activation of NF- κ B without degradation of I κ B (ref.18). Consistent with this, we found that overexpressed REDD1 interacted with I κ B α and promoted nuclear translocation of NF- κ B p65 in HEK293 cells (Fig, 6b, c), as well as stimulated nuclear NF- κ B p65 translocation without affecting I κ B α levels in 3T3-L1 cells (Supplementary Figs. 8e, f, and 9e). We have described these results in the Results section and added the following sentence in the Discussion section. “REDD1 overexpression is sufficient for NF- κ B activation and induction of cytokine gene expression through a mechanism in which REDD1 directly interacts with I κ B α , without altering IKK β phosphorylation and I κ B α degradation¹⁸, suggesting that REDD1 stimulates atypical NF- κ B activation by interactively sequestering I κ B α rather than degrading it.”

15. In Figure 6 d, c, expression of REDD1 and *Redd1*^{KKAA} must be shown to be sure that the level of expression is the same.

Response: As per the reviewer’s suggestion, we performed western blotting for determining REDD1 levels in 3T3-L1 cells infected with Ad-control, Ad-*Redd1*, or *Redd1*^{KKAA} and found no different expression levels of REDD1 between the cells transfected with adenovirus containing WT and *Redd1*^{KKAA}. Data was added to Supplementary Fig. 9e.

16. In Figure 7, what is the expression level of REDD1^{KKAA} in HFD mice? as well as NF- κ B activity, I κ B α expression and mTORC1 activity?

Response: We found that the expression level of REDD1 was not different in the eWAT, liver, and skeletal muscle between *Redd1*^{KKAA} mice and their littermates. The results were added to Supplementary Fig. 1e. We also found that *Redd1*^{KKAA} mice had reduced the HFD-induced increase in NF- κ B activity without affecting I κ B α expression in eWAT compared to that in WT mice. We have added these results to Fig. 7e and Supplementary Fig. 10i, j. Moreover, the mTORC1 downstream signaling mediators

S6K and S6 phosphorylation were increased in *Redd1^{KKAA}* mice and their control littermates on feeding with HFD, but the increase was not different between the two groups. These results were added to Supplementary Fig. 12c. In addition, we have added a new paragraph to the Discussion section as mentioned in the response to comment #7.

Reviewer #2 (Remarks to the Author):

Completely fails to attract readers due to lack of novelty. See for the reference. “Sci Rep. 2017 Aug 1;7(1):7023. doi: 10.1038/s41598-017-07182-z. Implication of REDD1 in the activation of inflammatory pathways.” Authors fail to show raw data instead of fair data in source of files. The manuscript is full of severe major inaccuracies and is not suitable for nature communications.

Response: According to the editorial decision, we have not responded to the concerns of Review #2 regarding novelty. However, we have now provided raw data of our experiments in Source data.

Reviewer #3 (Remarks to the Author):

In this manuscript, the authors investigated the role of REDD1 in obesity and metabolic disorders. The authors utilized several different mouse models including whole body knock out, and conditional knockout mice to study how Redd1 in different cell types regulates obesity and insulin resistance. For example, they found that adipose specific Redd1 knock out alleviated obesity, meta-inflammation, macrophage infiltration, and insulin resistance. Myeloid specific Redd1 knockout only alleviated macrophage infiltration, meta-inflammation, and insulin resistance but not obesity/adiposeness. Additionally, the authors demonstrate possible interactions with Lys219/220 on Redd1 and Glu85 and Asp73 on I κ B α . The animal experiments seem to be well executed and their findings may increase our understanding of the function of REDD1 in obesity and metabolic disorders. However, the study is relatively descriptive and lack mechanistic insight. The interaction between Redd and I κ B α cannot fully explain the phenotype of the mice. Further, adiponectin-cre only deleted REDD1 in mature adipocyte but not in adipose progenitors, deficiency of adipocyte REDD1 should not affect adipocyte differentiation. Thus, the conclusion and the proposed model in Extended Data Fig 12 is not correct. I have the following specific comments and suggestions:

1. The authors studied adipocyte-specific REDD knockout mice and found that deficiency of adipocyte REDD1 inhibited diet-induced obesity and insulin resistance. However, the underlying mechanisms were not provided. Since adiponectin-cre only deleted REDD1 in mature adipocyte but not adipose progenitors, deficiency of adipocyte REDD1 should not affect adipocyte differentiation. So the conclusion and the proposed model in Extended Data Fig 12 is not correct. More detailed mechanistic studies should be performed to explain this phenotype.

Response: Adiponectin is expressed in adipocytes but not in SVF cells or adipocyte progenitors; however, it is expressed from the early phase of preadipocyte differentiation (ref.64). Therefore, *Redd1* is eliminated from the onset of adipocyte differentiation in *Redd1* ^{Δ Adipoq} mice after HFD feeding, progressively leading to

inefficient or delayed differentiation into mature adipocytes and prevention of HFD-induced obesity. Indeed, we found that *Redd1*^{-/-} and *Redd1*^{ΔAdipoq} mice showed similarly retarded weight gain after 6 weeks of HFD compared to that by the WT mice. Based on these results, we added the following paragraph to the Discussion section and modified the proposed model in Supplementary Fig. 13. “Since adiponectin expression is increased over the entire period from the early phase of preadipocyte differentiation to fully differentiated adipocytes⁶⁴, *Redd1* begins to be eliminated from the early differentiated adipocytes of *Redd1*^{ΔAdipoq} mice after HFD feeding, leading to inefficient or delayed differentiation into mature adipocytes and preventing HFD-induced obesity. This possibility was confirmed by the finding that *Redd1* deficiency or knockdown suppressed MDI-induced differentiation of SVF cells and 3T3-L1 preadipocytes. On the other hand, we found that *Redd1*^{ΔLysM} mice ameliorated HFD-induced inflammation and metabolic dysfunction without altering adipogenesis and weight gain. These findings provide definitive evidence that HFD-induced REDD1 stimulates obesity and insulin resistance through cell type-specific functions, such as adipocyte differentiation and macrophage inflammation, in adipose tissue through the atypical activation of NF-κB (Supplementary Fig. 13).”

2. The authors found that REDD1 can directly interact with IκBα but this may only explain the increased inflammation phenotype but not increased adipocyte inflammation. NF-κB signaling plays complex role in regulating adipocyte differentiation and NF-κB itself does not directly regulate *Pparg* and *Cebpa* transcription. The observed adipogenesis effects may not due to NF-κB's transcriptional activity but other proteins associated with NF-κB signaling (e.g. IKK, Wnt). More experiments on this topic should be performed.

Response: Several studies have reported that the NF-κB pathway promotes adipogenesis through upregulation of PPARγ and CEPBα by IKKβ-mediated β-catenin degradation or NF-κB-Smurf2-dependent β-catenin degradation, or NF-κB-mediated CEPBα expression (ref.11,23,26). Based on this concept, we conducted experiments to confirm these possibilities and found that the REDD1-NF-κB axis could directly

stimulate CEBP α expression independent of β -catenin degradation and there is a positive cross-regulation loop between *Cebpa* and *Pparg* expression (ref.28). Therefore, we have now added new data to Supplementary Fig. 8g-i and the following contents to the Results section. “Since the adipogenic genes, *Pparg* and *Cebpa*, are known to be upregulated by degradation of β -catenin through IKK β -mediated β -catenin phosphorylation or NF- κ B-induced Smurf2 expression^{11,23,26}, we examined the role of REDD1 in the expression of these genes. REDD1 overexpression increased PPAR γ and CEBP α levels without affecting IKK $\alpha\beta$ phosphorylation or nuclear β -catenin accumulation in 3T3-L1 cells (Supplementary Fig. 8g), indicating that REDD1 promotes adipogenesis independent of β -catenin degradation. Consistent with the previous study showing NF- κ B-dependent transcription of CEBP α ²⁷, six putative NF- κ B binding sites were predicted on *Cebpa* promoter using the ALGGEN PROMO software v8.3. (<http://alggen.lsi.upc.es>). Among them, the proximal site centered at -1052 bp had higher transcription activity than others, which was confirmed in *Redd1*-overexpressing cells using chromatin immunoprecipitation (ChIP) assay and promoter activity analysis (Supplementary Fig. 8h, i). This suggests that REDD1-induced NF- κ B activation increases preadipocyte differentiation by transcriptional upregulation of *Cebpa* and in turn the positive cross-regulation loop between *Cebpa* and *Pparg* expression²⁸. However, further detailed function of each site needs to be analyzed.”

3. The authors mentioned that mice were on HFD for 10 or 16 weeks. Please specify the groups that were placed for duration of HFD. What’s the reason for different feeding duration for the studies?

Response: In general, adipogenic genes are induced in the early stages of adipocyte differentiation and adipogenesis. Thus, mice were fed HFD for 10 and 16 weeks to examine adipogenic gene expression and obesity-induced metabolic complications, respectively. We have now modified the sentence as follows: “Six-week-old male mice were fed either NC (10% calories as fat, #D12450B, JA BIO, Inc., Suwon, South Korea) or HFD (60% calories as fat, #D12492, JA BIO, Inc.) for 10 weeks to analyze expression levels of adipogenic genes *Pparg*, *Cebpa*, and *aP2* or for 16 weeks to

examine obesity and metabolic complications.”

4. The authors provided data for insulin signaling in liver and skeletal muscle. Since obesity is a major focus of this paper, the authors should also include some insulin signaling data in adipose tissue.

Response: As per the reviewer’s suggestion, we examined the insulin signaling in adipose tissue of *Redd1*^{-/-}, *Redd1*^{ΔAdipoq}, *Redd1*^{ΔAdipoq}, *Redd1*^{KKAA} and control (WT and *Redd1*^{fl/fl}) mice fed NC or HFD and found that insulin signaling was rescued in adipose tissues similarly to that in the liver and skeletal muscle of HFD-fed *Redd1*^{-/-}, *Redd1*^{ΔAdipoq}, and *Redd1*^{ΔLysM} mice, but not in NC-fed mouse groups. We have now added these results to Figures 2f, 3i, 4i, and 7j.

5. The authors state (line 354) that REDD1-induced MCP-1 production in adipocytes is an important player in macrophage activation. Additionally, results show REDD1 signaling in macrophages can similarly induce pro-inflammatory activation.

Response: As demonstrated in a previous study REDD1 overexpression promotes the expression of inflammatory genes, such as iNOS and TNF- α , in macrophage cell line RAW264.7 via atypical NF- κ B activation (ref.18). We also found that adenoviral overexpression of REDD1 stimulates NF- κ B activation and expression of inflammatory genes, such as TNF- α , IL-1 β , and IL6, in mouse peritoneal macrophages. Therefore, we have now added these results to Fig. 5f,g and Supplementary Fig. 8j and the following description to the Results section “In addition, REDD1 overexpression increased NF- κ B-driven luciferase activity as well as TNF- α , IL-1 β , and IL-6 production in mouse peritoneal macrophages (Fig. 5f, g and Supplementary Fig. 8j), which is consistent with the previous results demonstrating the proinflammatory action of REDD1 in RAW264.7 cells through atypical NF- κ B activation¹⁸.”

6. Previous data suggest IL-6 as a major player in regulating insulin signaling (Cai, 2005, Nature Medicine 11, 183-190). The authors should provide more discussion and

data related to IL-6, for example:

- a. Provide data for IL-6 levels in plasma and tissue from the *Redd1* ^{Δ LysM} model.
- b. Provide data for IL-6 target genes in different tissues (e.g. SOCS and Stat).
- c. More discussion on IL-6 role in insulin signaling and possibly move IL-6 data to main figures

Response: As recommended by the reviewer, (a) we have now added IL-6 plasma levels data in all animal models and in vitro cell culture systems to main Figures 1h, 3f, 4e, 5g, and 7g. (b) We also found that levels of pStat3 and *Socs3* increased in the eWAT, liver, and skeletal muscle of HFD-fed WT mice, levels of which were rescued in HFD-fed *Redd1*^{-/-} mice, but were not different in NC-fed mouse groups. These results are now added to the Supplementary Fig. 3b,c. (c) In addition, we have now discussed the functional importance of IL-6 in obesity-induced impairment of insulin signaling in the Discussion section as follows. “In addition, IL-6 plays an important role in negatively regulating insulin signaling through inhibition of IRS-1 and IRS-2 phosphorylation and/or promotion of ubiquitin-mediated IRS-1 and IRS-2 degradation by stimulating Stat3-dependent SOCS expression^{21,63}. Indeed, *Redd1*^{-/-} mice had reduced HFD-induced IL-6 production, Stat3 phosphorylation, and *Socs3* expression. These results suggest that HFD-induced REDD1 elicits meta-inflammation and subsequently impairs insulin-driven IRS–PI3K–Akt signaling.”

7. The authors should also rephrase some claims that are made in the results section to avoid misinterpretation. Since the authors chose to show both NC and HFD groups, they need to clarify which group they are referring to when making a claim. For example:

- a. Line 136: Be sure to state that the reduction in hepatic gluconeogenesis occurs in the HFD group.

Response: We have now added the experimental conditions of NC or HFD feeding to make syntax and sentences clear throughout the text.

8. In line with the previous comment, authors should consider placing NC data in supplemental data and moving some data (such as IL-6 and others) to the main text.

Response: As recommended, we have moved all the NC data to Supplementary Figures except for the experimental results involving *Redd1*^{-/-} mice, as they are necessary for comparing the results obtained from NC- and HFD-fed *Redd1*^{-/-} mice.

9. The authors should also provide data to verify their KO models (e.g. by western blotting, QPCR, or citations).

Response: *Redd1*^{-/-} mice were generated and used in our previous study (ref.42), and detailed information is available in our published article (ref.42) and is also provided in the Methods section. In addition, REDD1 deletion was further demonstrated in the eWAT, liver, and skeletal muscle of HFD-fed *Redd1*^{-/-} mice by western blotting, and the related data is added to Supplementary Fig. 1c. We have also provided in detail information on the generation and verification of *Redd1*^{fl/fl} and *Redd1*^{KKAA} mice in Supplementary Figs. 13 and 15. Finally, we verified REDD1 expression levels in the eWAT, liver, and skeletal muscle of *Redd1*^{ΔAdipoq} and *Redd1*^{KKAA} mice and have presented the results in Supplementary Fig. 1d, e. Additional information is now added to the Methods section.

Reviewer #4 (Remarks to the Author):

I have read the manuscript NCOMMS-21-39925 entitled "The REDD1–NF- κ B axis is crucial for adipogenesis, meta-inflammation, and metabolic dysfunction", submitted to Nature Comments for publication. As requested, I will address my main contribution to molecular docking and molecular dynamics aspects. The computational data presented in the manuscript is highly speculative, at least in how it was introduced here. Therefore, other assays are necessary to make outcomes more reliable.

Molecular docking is a powerful computational technique for finding binding poses and estimating binding free energy between two molecular entities. However, it is highly recommendable to use more than one program to accomplish this task. The authors limited themselves to using only the HADDOCK server, a well-known protein-protein server. In this case, the solution is highly dependent on the residues defined as active.

On the other hand, the best pose may not be unique, and other possibilities must be considered. The list of protein-protein docking servers is long. (Examples are: The ClusPro web server for protein-protein docking, ZDOCK, Hex Protein Docking, to name a few). These other servers could retrieve equally valid poses that might be potential solutions to the complex formation. Complementary analyses such as calculating the electrostatic potential map for both proteins, *Redd1* and I κ B α , to investigate complementarity would reinforce the quality of the results.

Furthermore, there is no quantitative estimation of the binding intensity. No measure of the interaction energy or even a rough estimate of the binding ΔG using a server as PISA (or equivalent) is given. This would be of particular value when investigating docking between mutated *Redd1* protein and the receptor. This quantitative measure must be estimated for the wild and mutated complexes and have to be correlated with experimental data is possible.

Finally, molecular docking is necessary but is not sufficient if taken isolatedly. Although the authors claim they used molecular dynamics simulations in their work, no description of any simulation result was found in the manuscript. Moreover, no dynamic information is given. For example, there is no description of salt bridges

occupancies, or hydrogen bonds mean half-lives, increasing the results.

Furthermore, it is mandatory to monitor complex stability from docking pose. The system's dynamical evolution provides a clue on the pose's veracity. Badly docked molecules do not remain bound for longer times. Wrong poses will undo in times less than dozens of nanoseconds. So, RMSD of the complex, taking the initial conformation as a reference, will show whether the pose was kept over the simulation time. Simulations must be carried out in triplicates for longer times than the one simulated here. It is advisable to set three different simulations of about 100 ns minimum for each complex, given that the complex involves heavy macromolecular moieties.

Response: Thank you for your helpful review. As per your advice, we have made some improvements in our manuscript. It is well established that the protein-protein docking simulation results are in general not highly reliable. Therefore, as per your suggestion, we performed Protein-Protein docking using three different methods, HADDOCK, HDOCK, and ZDOCK, in order to obtain consensus results. Binding modes that were compatible with the experimental data were obtained from HADDOCK and HDOCK. These findings are included in the Results section (Supplementary Fig. 9). We are sorry for the method text regarding the MD. Previously, we performed MD simulation only for obtaining more stable/accurate binding modes over the protein-protein docking structure. Hence, we did not pay much attention to some parts of the MD analysis. We apologize for not explaining this well in the manuscript. In the present manuscript, we did not perform MD simulations in order to save time and effort. Therefore, in the current version, the MD part was completely removed from the methods.

Although you have suggested to do ESP analysis and MD simulations to check the binding mode stability etc., we believe that the focus of the study is experimentally oriented and that extensive molecular modeling will be beyond the scope for the manuscript.

We thank you again for your kind and detailed comments. They have been very helpful in improving our manuscripts.

REVIEWER COMMENTS

Reviewer #1 (Remarks to the Author):

The authors have adequately responded to most criticisms. Overall I think the paper provides important new insight into the role of REDD1 in metabolism and warrants publication.

Reviewer #3 (Remarks to the Author):

While The authors provided additional data to address some comments, they did not sufficiently address some key questions:

1. The authors used adipocyte-specific Redd1 KO mice (by using adiponectin-cre) and found that Redd1 Δ Adipoq mice also had decreased weight gain after HFD feeding as compared with control mice. They also claimed that deficiency of Redd1 in adipocytes decreased adipogenesis. However, multiple studies have confirmed that adiponectin-cre only targets mature or differentiated adipocytes but not progenitors in vivo (e.g., Jeffery e. et al., Adipocyte 2014 Jul 1;3(3):206-11). The reference #64 the authors cited is not convincing at all. To prove that the decreased obesity in Redd1 Δ Adipoq mice was due to decreased adipogenesis, the authors should isolate SVF and mature adipocytes from Redd1 Δ Adipoq mice to confirm that adiponectin-cre-mediated Redd1 deletion also occurs in SVF or progenitor cells, resulting in decreased SV cell differentiation.

2. The author did not show convincing in vivo adipogenesis data to support the mechanism for the decreased obesity in global or adipocyte-specific Redd1 KO mice. The adipogenesis results were mostly obtained from cultured SVF. To target Redd1 in adipocyte progenitors in vivo, the author should consider using appropriate Cre models (e.g., PdgfRa-cre). The decreased obesity in global or adipocyte-specific Redd1 KO mice could also due to changed energy expenditure but not adipogenesis. Did the authors measure energy expenditure in those mice? If these studies are not realistic for the current study, the author should discuss this topic in more details.

Reviewer #4 (Remarks to the Author):

All of my major concerns about molecular modeling have been addressed.

I agree with the authors in removing the MD section from this version of the text.

Response to reviewers' comments (2nd)

Reviewer #1 (Remarks to the Author):

Comment: The authors have adequately responded to most criticisms. Overall I think the paper provides important new insight into the role of REDD1 in metabolism and warrants publication.

Response: We appreciate the feedback provided by the Reviewer #1.

Reviewer #3 (Remarks to the Author):

While The authors provided additional data to address some comments, they did not sufficiently address some key questions:

Comment #1: The authors used adipocyte-specific *Redd1* KO mice (by using adiponectin-Cre) and found that *Redd1*^{ΔAdipoq} mice also had decreased weight gain after HFD feeding as compared with control mice. They also claimed that deficiency of *Redd1* in adipocytes decreased adipogenesis. However, multiple studies have confirmed that adiponectin-Cre only targets mature or differentiated adipocytes but not progenitors in vivo (e.g., Jeffery E. et al., *Adipocyte* 2014 Jul 1;3(3):206-11). The reference #64 the authors cited is not convincing at all. To prove that the decreased obesity in *Redd1*^{ΔAdipoq} mice was due to decreased adipogenesis, the authors should isolate SVF and mature adipocytes from *Redd1*^{ΔAdipoq} mice to confirm that adiponectin-Cre-mediated *Redd1* deletion also occurs in SVF or progenitor cells, resulting in decreased SV cell differentiation.

Response to comment #1: (1) As pointed out by the reviewer, we isolated or purified eWAT, SVF cells, adipocytes, liver tissues, and skeletal muscles from *Redd1*^{fl/fl} and *Redd1*^{ΔAdipoq} mice fed either NC or HFD and then examined the expression levels of REDD1. We found that REDD1 was specifically deleted in purified adipocytes but not in SVF cells or other organs. Therefore, we added these data to Supplementary Figure 1d and the following sentence to the Results section. “we examined the role of adipocyte REDD1 in HFD-induced adipogenesis and inflammation using *Redd1*^{ΔAdipoq} mice, in which REDD1 expression was specifically deleted in mature adipocytes but not in SVF cells, liver tissues, and skeletal muscles under HFD conditions (Supplementary Fig. 1d).”

(2) Next we examined whether REDD1 expression can be regulated during MDI-induced in vitro adipogenic differentiation of adipose SVF cells and compared the adipogenesis efficiency between MDI-induced 3T3-L1 cells and SVF cells isolated from eWATs obtained from WT, *Redd1*^{-/-}, and *Redd1*^{ΔAdipoq} mice. We found that SVF cells from *Redd1*^{ΔAdipoq} mice differentiated into adipocytes, similar to wild-type cells

when stimulated with MDI, but SVF cells from *Redd1*^{-/-} mice were inhibited. These findings were added to Figure 5c-f and Supplementary Figure 8a and the following reorganized paragraph to the Results section. “To investigate the functional role of REDD1 in adipogenesis, we compared the adipogenic potential of adipose SVF cells isolated from *Redd1*^{-/-} mice and their WT littermates. When cultured in a differentiation medium containing an adipogenic cocktail (MDI) of methylisobutylxanthine, dexamethasone, and insulin, SVF cells from WT mice presented effective induction of REDD1 expression from 12 h after the stimulation, followed by upregulation of PPAR γ and C/EBP α expression on day 2 of the stimulation (Supplementary Fig. 8a), suggesting that REDD1 is upregulated early in the adipogenic differentiation process. As expected, MDI-stimulated *Redd1*-deficient SVF cells showed poorer adipogenesis than WT cells (Fig. 5a and Supplementary Fig. 8b). In addition, *Redd1* knockdown by shRNA suppressed adipogenic differentiation of 3T3-L1 preadipocytes cultured in the differentiation medium (Fig. 5b and Supplementary Fig. 8c). These results suggest that REDD1 is important for adipogenic differentiation. On the other hand, the inhibition of adipogenic differentiation and *Pparg* and *Cebpa* mRNA expression in *Redd1* ^{Δ Adipoq} mice SVF cells upon MDI stimulation was insignificant (Fig. 5d). This phenomenon is likely due to the deletion of *Redd1* only in matured *Redd1* ^{Δ Adipoq} adipocytes, as shown by the unchanged levels of REDD1 on day 4 after MDI stimulation and the marked downregulation on day 8 (Fig. 5e), consistent with previous studies that adiponectin expression was restricted in mature adipocytes²⁴. However, lipogenic genes, such as acetyl-CoA carboxylase (*Acc*), fatty acid synthase (*Fasn*), and stearoyl-CoA desaturase-1 (*Scd-1*), were significantly downregulated in MDI-stimulated *Redd1* ^{Δ Adipoq} SVF cells (Fig. 5f). This suggests that REDD1 does not affect the differentiation of SVF cells into adipocytes in *Redd1* ^{Δ Adipoq} mice fed HFD but can inhibit fatty acid synthesis.”

(3) In addition, considering the above results, we changed the subtitle “Adipocyte *Redd1* deficiency reduces adipogenesis and meta-inflammation” to “Adipocyte *Redd1* deficiency reduces weight gain and meta-inflammation.”

(4) Finally, we removed Reference #64 and its related content. The final paragraph

of the Discussion section was modified as follows. “Taken together, our findings provide evidence that HFD-induced REDD1 stimulates obesity and insulin resistance through cell type-specific functions, such as adipogenesis and macrophage inflammation, in adipose tissue through the atypical activation of NF- κ B (Supplementary Fig. 13). REDD1-dependent NF- κ B activation was stimulated through IKK-independent NF- κ B activation by sequestering I κ B α , which masks the NLS of NF- κ B p65 and keeps it in an inactive state in the cytoplasm. Thus, we propose that the REDD1–NF- κ B axis is the molecular link between obesity-related inflammation and insulin resistance and should be targeted in new therapeutic strategies to normalize body weight and improve metabolic complications. Because REDD1-induced expression of proinflammatory cytokines, including TNF- α and IL-1, can subsequently promote the canonical NF- κ B pathway, the functional involvement of IKK β in HFD-induced adipogenesis, meta-inflammation, and glucose metabolism dysregulation cannot be ruled out⁷. We speculate that REDD1 is extremely important but not indispensable for promoting obesity-induced characteristics. Therefore, studies confirming the crosstalk between REDD1-dependent atypical and IKK β -dependent canonical mechanisms of NF- κ B activation in the pathogenesis of obesity-related metabolic complications are warranted.”

Comment #2: The author did not show convincing in vivo adipogenesis data to support the mechanism for the decreased obesity in global or adipocyte-specific Redd1 KO mice. The adipogenesis results were mostly obtained from cultured SVF. To target Redd1 in adipocyte progenitors in vivo, the author should consider using appropriate Cre models (e.g., PdgfRa-Cre). The decreased obesity in global or adipocyte-specific Redd1 KO mice could also due to changed energy expenditure but not adipogenesis. Did the authors measure energy expenditure in those mice? If these studies are not realistic for the current study, the author should discuss this topic in more details.

Response to comment #2: As suggested by the reviewer, in the Discussion section, we have described the possible mechanisms by which *Redd1* ^{Δ Adipoq} mice, in

which REDD1 was deleted in mature adipocytes as previously reported by Jeffery E. et al. (doi: 10.4161/adip.29674), reduce obesity-induced pathological characteristics, similar to *Redd1^{-/-}* mice. This phenomenon could be related to alterations in lipogenesis, lipolysis, or energy expenditure. To elucidate the *in vivo* preadipocyte-specific function of REDD1 in adipogenesis and obesity, further studies related to adipogenesis, obesogenesis, lipid metabolism, and energy expenditure using preadipocyte-specific *Redd1*-deficient (*Redd1^{ΔPdgfRa}*) mice are needed. Thus, we discuss these points in the revised manuscript as follows. “Notably, we found that the development of HFD-induced obesity was prevented in *Redd1^{ΔAdipoq}* mice; however, their SVF cells did not affect adipogenic gene expression and adipogenesis *in vitro* but inhibited lipogenic gene expression. This suggests that *Redd1* deletion in mature adipocytes of *Redd1^{ΔAdipoq}* mice, as previously reported²⁴, does not affect adipocyte differentiation (including mitotic clonal expansion) but can inhibit fatty acid synthesis (adipocyte hypertrophy). Adipocyte hypertrophy or lipohypertrophy, which is essential for adipose tissue growth and weight gain, results from not only *de novo* fatty acid synthesis but also impaired lipolysis or energy expenditure. The reduced WAT mass and body weight in *Redd1^{ΔAdipoq}* mice may be associated with decreased fatty acid synthesis or increased lipolysis or energy expenditure, consistent with previous studies that genetic inhibition of the NF-κB pathway increased fatty acid catabolism and energy expenditure in a mouse model of HFD-induced obesity^{25,48}. Therefore, the function of the preadipocyte-specific REDD1/NF-κB pathway in adipogenesis, lipid metabolism, and energy expenditure, should be investigated in more detail in a mouse model of diet-induced obesity using adipocyte progenitor cell-specific *Redd1*-deleted mice, such as *Redd1^{ΔPdgfRa}* mice, as previously reported²⁴.”

Reviewer #4 (Remarks to the Author):

Comment: All of my major concerns about molecular modeling have been addressed. I agree with the authors in removing the MD section from this version of the text.

Response to comment: We appreciate the feedback provided by the Reviewer #4.

REVIEWERS' COMMENTS

Reviewer #3 (Remarks to the Author):

The authors have conducted additional experiments to confirm that REDD1 is only deleted in mature adipocytes but not progenitors in their *Redd1 Δ Adipoq* mice. These results are consistent with many previous reports in the field. They have revised their manuscript and title accordingly and their discussions/conclusions are now appropriate to explain the mouse phenotypes. I appreciate the authors' efforts to clarify this important issue and to avoid publish wrong statements/conclusions that may warrant corrections/amendments in the future. Therefore, they have sufficiently addressed my comments and the revised manuscript is now suitable for publication in Nature Communications.

Response to reviewers' comments (3rd)

Reviewer #3 (Remarks to the Author):

Comment: The authors have conducted additional experiments to confirm that REDD1 is only deleted in mature adipocytes but not progenitors in their *Redd1^{ΔAdipoq}* mice. These results are consistent with many previous reports in the field. They have revised their manuscript and title accordingly and their discussions/conclusions are now appropriate to explain the mouse phenotypes. I appreciate the authors' efforts to clarify this important issue and to avoid publish wrong statements/conclusions that may warrant corrections/amendments in the future. Therefore, they have sufficiently addressed my comments and the revised manuscript is now suitable for publication in Nature Communications..

Response: We appreciate the feedback provided by the Reviewer #3.